# Visualization of accessible cholesterol using a GRAM domain-based biosensor

Dylan Hong Zheng Koh [1], Tomoki Naito [1], Minyoung Na[1], Yee Jie Yeap [1], Pritisha Rozario[1], Franklin L. Zhong [1,2], Kah-Leong Lim[1,3] & Yasunori Saheki [1,4] ✉

Cholesterol is important for membrane integrity and cell signaling, and dysregulation of the distribution of cellular cholesterol is associated with numerous diseases, including neurodegenerative disorders. While regulated transport of a specific pool of cholesterol, known as "accessible cholesterol", contributes to the maintenance of cellular cholesterol distribution and homeostasis, tools to monitor accessible cholesterol in live cells remain limited. Here, we engineer a highly sensitive accessible cholesterol biosensor by taking advantage of the cholesterol-sensing element (the GRAM domain) of an evolutionarily conserved lipid transfer protein, GRAMD1b. Using this cholesterol biosensor, which we call GRAM-W, we successfully visualize in real time the distribution of accessible cholesterol in many different cell types, including human keratinocytes and iPSC-derived neurons, and show differential dependencies on cholesterol biosynthesis and uptake for maintaining levels of accessible cholesterol. Furthermore, we combine GRAM-W with a dimerization-dependent fluorescent protein (ddFP) and establish a strategy for the ultrasensitive detection of accessible plasma membrane cholesterol. These tools will allow us to obtain important insights into the molecular mechanisms by which the distribution of cellular cholesterol is regulated.

Cholesterol plays an essential role in the structural integrity of cellular membranes and in cell signaling[1]. Dysregulation of cholesterol metabolism has been implicated in numerous disorders, including cardiovascular and neurological diseases[2–7]. Cells either acquire cholesterol from the extracellular environment via uptake of lipoproteins or synthesize it in the endoplasmic reticulum (ER)[3,8]. Regardless of the source, cholesterol is then delivered to the plasma membrane (PM), where it contributes to up to 40% of the total lipids in this bilayer[1,9–14].

Non-vesicular transport of cholesterol facilitated by lipid transfer proteins (LTPs) plays a major role in distributing cellular cholesterol[15–28]. Availability of membrane cholesterol for LTP-mediated non-vesicular transport is tightly regulated by the dynamic interactions between cholesterol and the membrane environment[9,29–32]. At steady state, the

majority of membrane cholesterol is inaccessible to LTPs due to sequestration via its complex formation with neighboring membrane lipids, such as phospholipids and sphingomyelin. Thus, only a small fraction of membrane cholesterol is accessible (also known as chemically active) for LTP-mediated non-vesicular transport at steady state[32–40]. Growing evidence suggests that accessible cholesterol plays critical roles in a range of cellular processes, including the regulation of hedgehog signaling and defense against pathogens[41–47]. Thus, there is great interest in the field to develop an effective approach for visualizing the distribution of accessible cholesterol in live cells.

Over the years, a number of probes have been developed to detect cholesterol in cellular membranes. They include bacteria-derived molecules such as filipin, a fluorescent polyene macrolide secreted by

[1]Lee Kong Chian School of Medicine, Nanyang Technological University, Singapore 308232, Singapore. [2]Skin Research Institute of Singapore (SRIS), Singapore 308232, Singapore. [3]National Neuroscience Institute, Singapore 308433, Singapore. [4]Institute of Resource Development and Analysis, Kumamoto University, Kumamoto 860-0811, Japan. ✉e-mail: yasunori.saheki@ntu.edu.sg

the bacteria *Streptomyces filipinensis*[48,49], Perfringolysin O (PFO), a cholesterol-dependent cytolysin (CDC) released by *Clostridium perfringens*, and Anthrolysin O (ALO), a CDC closely related to PFO that is released by *Bacillus anthracis*[50–52]. Filipin requires excitation by ultraviolet light and is easily photobleached. Thus, it is not commonly used for imaging live cells[53–55]. An increasing number of studies have utilized Domain 4 of PFO (PFO-D4) or ALO (ALO-D4) – Domain 4 is the C-terminal domain of the CDC responsible for cholesterol binding – as a probe to detect accessible PM cholesterol in live cells. For example, extracellularly applied recombinant PFO-D4 or ALO-D4 proteins have been successfully used to detect levels of accessible cholesterol in the PM as they bind strongly to the PM[35,44,47,56–58]. However, when PFO-D4 is expressed in the cytosol it binds less efficiently to the PM, possibly due to the large amount of anionic lipids present in the cytosolic leaflet of the PM[59]. In vitro, D4 binds less efficiently to membranes containing anionic lipids compared to membranes containing neutral lipids[60–62]. While substitution of aspartic acid at the 434th position of PFO-D4 to serine (D434S) (i.e., PFO-D4H) generally improves the binding of cytosolic PFO-D4 to the PM[63], even PFO-D4H fails to bind to the PM in some cell types at steady state[62,64–67]. Further, both ALO-D4 and PFO-D4H are known to inhibit the movement of accessible cholesterol[68–70]. Thus, there is a need for alternative biosensors that are more suitable for tracking accessible cholesterol in live cells. A potential candidate is the GRAM domain of GRAMD1s/Asters, which is a co-incidence detector for accessible cholesterol and anionic lipids[56,57,68]. ER-anchored GRAMD1s (GRAMD1a, GRAMD1b, GRAMD1c) sense the elevation of accessible cholesterol in various cellular membranes, including the PM and Golgi, via their GRAM domain and transport accessible cholesterol to the ER via their StART-like domain, contributing to the maintenance of cellular cholesterol homeostasis[28,57,68,71–75]. While the GRAM domain only weakly binds to the PM at steady state, it binds strongly to the PM when there is a transient expansion of the accessible pool of cholesterol in the PM[57,68,71]. Thus, we hypothesized that mutating a critical residue of the GRAM domain that is responsible for accessible cholesterol detection sensitivity would allow us to detect accessible cholesterol in the PM (and possibly other cellular membranes) even at steady state.

Here, we identify a mutant of the GRAMD1b GRAM domain (GRAM-W) that allows visualization of accessible PM cholesterol at steady state in live cells. GRAM-W additionally detects accessible cholesterol derived from low-density lipoproteins (LDLs) in lysosomal membranes. We use GRAM-W to successfully visualize accessible cholesterol in a number of different cell types, including human immortalized keratinocytes and induced pluripotent stem cells (iPSC)-derived dopaminergic neurons. Inhibiting de novo cholesterol biosynthesis largely reduces the binding of GRAM-W to the PM in keratinocytes, but not other cell types examined. This reveals differential dependencies on cholesterol biosynthesis and uptake for maintaining levels of accessible PM cholesterol in different cell types. Finally, we combine GRAM-W with a dimerization-dependent fluorescent protein (ddFP)[76,77] and show that ddFP-tagged GRAM-W allows real-time visualization of accessible cholesterol at the PM with unprecedented sensitivity. These tools will allow us to better understand the molecular mechanisms that are responsible for the maintenance of accessible cholesterol in cellular membranes.

## Results

### Visualization of accessible cholesterol by a GRAM-W biosensor

The GRAM domain of GRAMD1b (GRAM_{1b}) acts as a co-incidence detector of accessible cholesterol and anionic lipids, including phosphatidylserine (PS)[57,68]. Because of this property, GRAM_{1b} binds to cellular membranes that are enriched in PS, such as the PM, when levels of accessible cholesterol are elevated. We previously demonstrated that the glycine at the 187th position of GRAM_{1b} (G187) is key for determining its sensitivity to accessible cholesterol (Fig. 1a)[68].

The wild-type GRAM domain is recruited to the PM only when there is a sufficient increase in accessible PM cholesterol[57,68,71]. To identify a variant of GRAM_{1b} that binds to the PM even at steady state, we sought to increase its sensitivity to accessible cholesterol by mutating G187. We changed G187 to every other amino acid in an EGFP-tagged version of GRAM_{1b} and expressed each of these variants in HeLa cells. Recruitment to the PM was assessed using spinning disk confocal (SDC) microscopy and line scan analysis (Fig. 1b, c and Supplementary Fig. 1a). Recruitment of EGFP-GRAM_{1b} to the PM was generally increased by replacing G187 with a hydrophobic residue such as phenylalanine (F), methionine (M), isoleucine (I), or leucine (L). Strikingly, replacing G187 with tryptophan (W) dramatically enhanced the binding of EGFP-GRAM_{1b} to the PM (Fig. 1b, c and Supplementary Fig. 1a), indicating that the GRAM_{1b} variant with the G187W mutation (hereafter referred to as GRAM-W) detects accessible PM cholesterol with superior sensitivity compared to other variants.

To characterize the co-incidence detection properties of GRAM-W, we purified wild-type (WT) GRAM_{1b}, a variant of GRAM_{1b} carrying the G187L mutation (GRAM-H), which we previously characterized[68], and GRAM-W recombinant proteins. We then compared their abilities to bind liposomal membranes containing a fixed amount of PS (20%) and increasing amounts of cholesterol (0–50%). The binding curve associated with GRAM-W was shifted toward lower levels of cholesterol compared to those of WT GRAM_{1b} or GRAM-H (Fig. 1d). This indicated that GRAM-W was more sensitive to membranes containing both cholesterol and PS but it remained unclear whether the G187W mutation increased sensitivity of GRAM_{1b} to cholesterol, PS, or both. To disentangle these possibilities, we generated liposomes containing increasing amounts of cholesterol (0–60%) (Fig. 1e) or PS (0–80%) (Fig. 1f), and examined the binding efficiencies of purified GRAM_{1b} protein variants to these liposomes. GRAM-W bound to liposomes containing cholesterol more strongly than WT GRAM_{1b} or GRAM-H. For liposomes that contained 60% cholesterol, ~80% of GRAM-W bound to these liposomes, whereas only ~50% of GRAM-H and ~20% of WT GRAM_{1b} bound (Fig. 1e). Interestingly, GRAM-W also bound more strongly to liposomes containing PS than seen with WT GRAM_{1b} or GRAM-H. For liposomes that contained 60% PS, ~60% of GRAM-W bound, whereas only ~20% of both WT GRAM_{1b} and GRAM-H bound (Fig. 1f). These results are consistent with GRAM-W exhibiting increased sensitivity to both cholesterol and PS.

To validate the ability of GRAM-W to sense cholesterol in the PM, HeLa cells expressing EGFP-GRAM_{1b} WT, EGFP-GRAM-H, or EGFP-GRAM-W were depleted of cholesterol by treating them with mevastatin and lipoprotein-deficient serum (LPDS) for 16 hours (a condition that prevents de novo cholesterol biosynthesis and LDL uptake)[33,57,68]. This treatment significantly decreased the ability of both EGFP-GRAM-H and EGFP-GRAM-W to bind the PM (EGFP-GRAM_{1b} WT was not bound to the PM even before treatment) (Fig. 1g, h). Further, treating these cells with methyl-β-cyclodextrin (MCD), which removes accessible cholesterol from the PM[78–80], also reduced the PM binding of these proteins (Supplementary Fig. 1b, c). MCD treatment did not affect the levels of major anionic lipids in the PM, including PS, phosphatidylinositol 4-bisphosphate (PI4P), and phosphatidylinositol 4,5-bisphosphate [PI(4,5)P_2], as assessed by mCherry-tagged Lact-C2 (mCherry-Lact-C2), iRFP-tagged P4M (iRFP-P4M), and iRFP-tagged PH domain from PLCδ1 (iRFP-PH-PLCδ1), respectively (Supplementary Fig. 1d, e). Therefore, reduced binding of EGFP-GRAM-W to the PM of MCD-treated cells did not result from changes in the levels of anionic lipids in the PM. The dissociation of GRAM-W from the PM was also examined in real time using total internal reflection fluorescence (TIRF) microscopy Fig. 1i, j). Both EGFP-GRAM-H and EGFP-GRAM-W dissociated from the PM within 15 min of MCD treatment. These results together confirm that GRAM-W is sensitive to accessible cholesterol in the PM.

A transient expansion in the accessible pool of PM cholesterol, either via the liberation of inaccessible cholesterol from sequestration

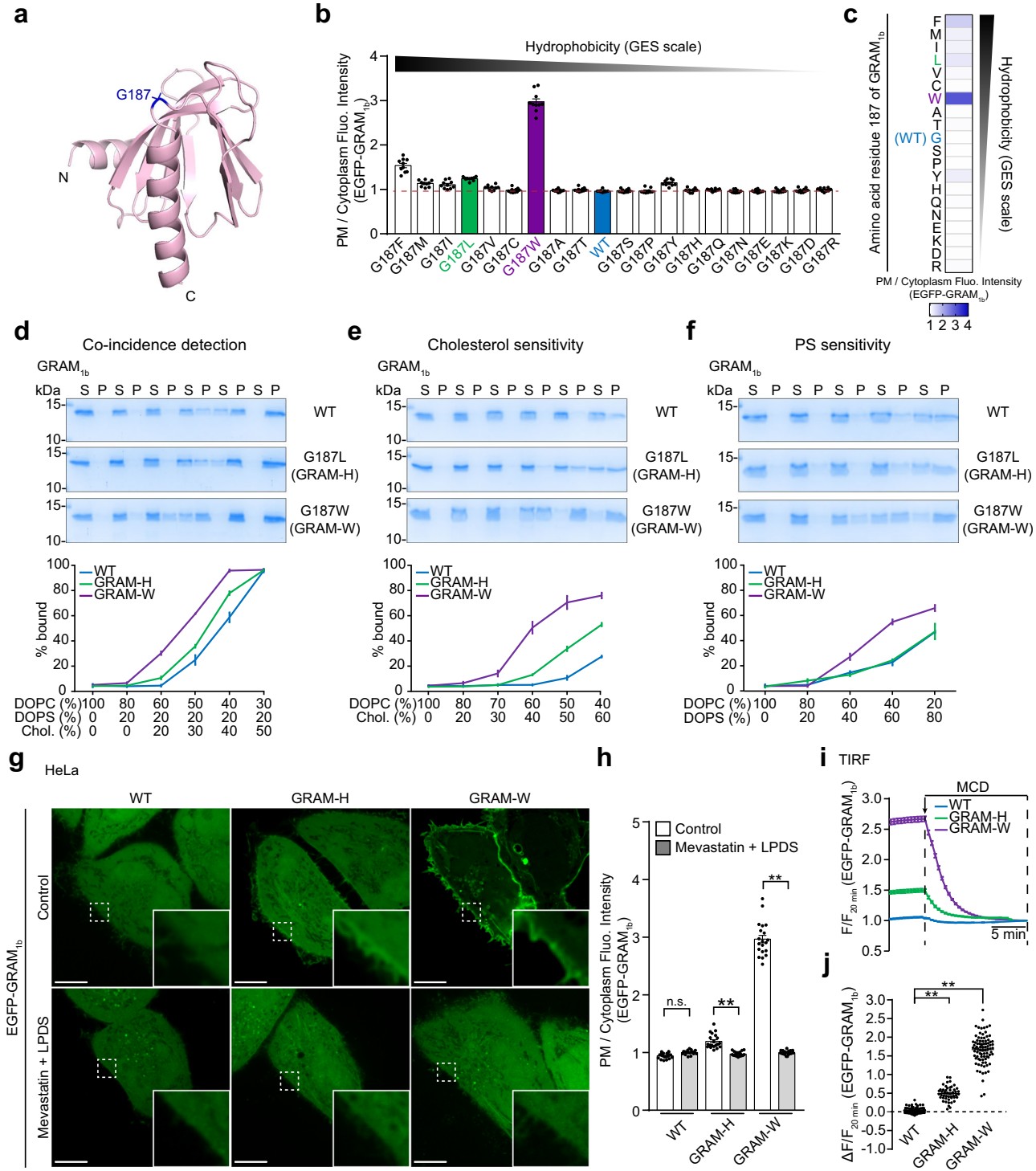

or via an increase in cholesterol content, is followed by rapid transport of accessible cholesterol from the PM to the ER via cholesterol-harboring LTPs such as GRAMD1s. This transport suppresses the activity of the transcription factor, SREBP-2, which functions as the master regulator of cholesterol biosynthesis and uptake[57,71,81–85], and activates acyl-coenzyme A (CoA):cholesterol acyltransferase (ACAT) in the ER for cholesterol esterification and lipid droplet formation[86–91]. To rule out the possibility that GRAM-W inhibits the transport of accessible cholesterol from the PM to the ER, we expressed EGFP-GRAM-W stably and monitored SREBP-2 cleavage, whose suppression indicates SREBP-2 inactivation, as well as the formation of lipid droplets. HeLa cells stably expressing EGFP-GRAM-W were generated via lentiviral

transduction. To induce the suppression of SREBP-2 cleavage, cells were treated with sphingomyelinase (SMase) to transiently increase accessible PM cholesterol (by liberating inaccessible PM cholesterol from sphingomyelin-mediated sequestration)[33,57,92,93]. The status of SREBP-2 cleavage was assessed before and after SMase treatment by immunoblotting. In control HeLa cells, SMase treatment resulted in less SREBP-2 cleavage, consistent with the suppression of SREBP-2 activity. A similar effect was seen in HeLa cells stably expressing EGFP-GRAM-W (Supplementary Fig. 2a, b). To monitor the formation of lipid droplets, cholesterol was loaded to the PM using the complex of MCD and cholesterol (MCD/Cholesterol). The formation of lipid droplets was assessed before and after MCD/Cholesterol treatment by

**Fig. 1 | Visualization of accessible cholesterol by a GRAM-W biosensor. a** The ribbon diagram of the modelled GRAM domain of GRAMD1b (GRAM$_{1b}$). The cholesterol-sensing amino acid, G187, is highlighted in blue. N, N-terminus; C, C-terminus. **b** Quantification of the ratio of PM signals to the cytosolic signals of wild-type EGFP-tagged GRAM$_{1b}$ (EGFP-GRAM$_{1b}$) (WT) and mutant versions of EGFP-GRAM$_{1b}$, as assessed by confocal microscopy and line scan analysis of HeLa cells (mean ± SEM, $n = 10$ cells from one experiment for all conditions). A red dashed line indicates the average value of WT. Amino acids are ranked according to Goldman-Engelman-Steitz (GES) hydrophobicity scale. **c** Heatmap representation of the quantified data as shown in **b. d–f** Liposome sedimentation assays of wild-type GRAM$_{1b}$ proteins (WT), and mutant versions of GRAM$_{1b}$ proteins carrying either G187L (GRAM-H) or G187W (GRAM-W) mutation (mean ± SEM, $n = 3$ independent experiments for all conditions). Bound proteins [pellet, (P)]; Unbound proteins [supernatant, (S)]. DOPC, 1,2-dioleoyl-sn-glycero-3-phosphocholine; DOPS, 1,2-dioleoyl-sn-glycero-3-phospho-L-serine; Chol., cholesterol. **g** Confocal images of live HeLa cells expressing either wild-type EGFP-GRAM$_{1b}$ (WT), EGFP-GRAM-H

(GRAM-H), or EGFP-GRAM-W (GRAM-W). Cells were incubated in either control medium or medium supplemented with mevastatin (50 μM) and 10% lipoprotein-deficient serum (LPDS) for 16 hrs to deplete cholesterol before imaging. Insets show at higher magnification the regions indicated by white dashed boxes. Scale bars, 10 μm. **h** Quantification of the ratio of PM signals to cytosolic signals of EGFP-GRAM$_{1b}$, as assessed by confocal microscopy and line scan analysis of HeLa cells expressing indicated constructs as shown in **g** (mean ± SEM, $n = 20$ cells for all conditions; data are pooled from two independent experiments; two-tailed unpaired Student's $t$ test, **$P < 0.0001$. n.s. denotes not significant). **i** Time course of normalized EGFP-GRAM$_{1b}$ signals, as assessed by TIRF microscopy, from HeLa cells expressing indicated constructs. Methyl-β-cyclodextrin (MCD) treatment (5 mM) is indicated. **j** Values of $\Delta F/F_{20\,min}$ corresponding to the timepoint as indicated by the arrow in **i** [mean ± SEM, $n = 92$ cells (WT), $n = 47$ cells (GRAM-H), $n = 87$ cells (GRAM-W); data are pooled from three independent experiments for all conditions; Dunnett's multiple comparisons test, **$P < 0.0001$].

LipidTOX staining. In control HeLa cells, MCD/Cholesterol treatment resulted in significant increase in the number and size of lipid droplets (Supplementary Fig. 2c–e). Importantly, such lipid droplet formation was largely suppressed by ACAT inhibitor, SZ58-035, showing that the formation of lipid droplets upon cholesterol loading to the PM is dependent on cholesterol esterification in the ER by ACAT (Supplementary Fig. 2c–e). A similar formation of lipid droplets was seen in HeLa cells stably expressing EGFP-GRAM-W, and it was also suppressed by SZ58-035 (Supplementary Fig. 2c–e). These results support that the expression of GRAM-W does not interfere with the extraction/transport of accessible cholesterol by intracellular cholesterol transport machineries, allowing the effective visualization of accessible cholesterol distribution in cells.

## GRAM-W detects LDL-derived accessible cholesterol in lysosomal membranes and the PM

Cells acquire cholesterol through the uptake of lipoproteins, primarily in the form of LDLs[8,9,94,95]. LDLs are delivered to lysosomes via receptor-mediated endocytosis, and then LDL-derived cholesterol is exported from lysosomes and delivered to other cellular compartments[3,8,95,96]. In steady-state HeLa cells, vesicular structures reminiscent of lysosomes were decorated by EGFP-GRAM-W, suggesting that GRAM-W may also detect accessible cholesterol in lysosomal membranes (Fig. 1g). HeLa cells stably expressing EGFP-GRAM-W were transiently transfected with LAMP1-miRFP (a lysosomal marker) and imaged under SDC-structured illumination microscopy (SDC-SIM). A small fraction of LAMP1-miRFP-labelled lysosomes were indeed decorated by EGFP-GRAM-W, indicating that EGFP-GRAM-W can detect accessible cholesterol on lysosomal membranes, in addition to the PM (Fig. 2a). Starving cells with mevastatin and LPDS for 16 hours reduced the level of EGFP-GRAM-W bound to lysosomal membranes (Fig. 2a), consistent with the ability of EGFP-GRAM-W to detect accessible cholesterol in cellular membranes. Furthermore, inhibiting Niemann-Pick disease type C1 (NPC1), a protein required for egress of LDL-derived cholesterol from lysosomes[97], by U18666A[98] for 16 hours resulted in reduction in the level of EGFP-GRAM-W bound to lysosomal membranes as well as the PM (Fig. 2a). These results suggest that NPC1, whose dysfunction is associated with neurodegeneration[8], plays an important role in increasing the levels of accessible cholesterol in lysosomal membranes and contributes to the maintenance of intracellular cholesterol distribution.

To date, there is no effective approach for tracking the movement of accessible cholesterol from one cellular compartment to another in live cells. Based on the ability of GRAM-W to detect accessible cholesterol in various cellular membranes, we reasoned it may serve as a reliable probe for tracking LDL-derived accessible cholesterol. HeLa cells stably expressing EGFP-GRAM-W were starved using mevastatin and LPDS for 16 hours and supplemented with LDL. During this time,

LDL-derived accessible cholesterol was tracked using time-lapse SDC imaging. LDL supplementation resulted in the gradual recruitment of EGFP-GRAM-W to the PM in addition to the lysosomal membranes over a 12-hour imaging period (Fig. 2b–d, Supplementary Fig. 3a and Supplementary Movie 1). By contrast, there was no significant recruitment of WT EGFP-GRAM$_{1b}$ to the PM under these same conditions (Fig. 2b–d and Supplementary Movie 1). The binding of EGFP-GRAM-W to the PM was compared with that of a well-establish accessible cholesterol biosensor based on PFO-D4 carrying the D434S mutation (D4H) [i.e., purified recombinant mCherry-tagged D4H (mCherry-D4H) protein] to the PM. HeLa cells stably expressing EGFP-GRAM-W were starved using mevastatin and LPDS for 16 hours and supplemented with LDL for different periods of time. They were subsequently incubated with purified recombinant mCherry-D4H protein for 10 min. The binding of mCherry-D4H proteins to the extracellular leaflet of the PM and the binding of EGFP-GRAM-W to the cytosolic leaflet of the PM were then simultaneously analyzed by SDC microscopy (Supplementary Fig. 3b–d). Overall, the timing of the increase in the binding of EGFP-GRAM-W to the cytosolic leaflet of the PM was similar to that of mCherry-D4H proteins to the extracellular leaflet of the PM. Increase in the PM binding of EGFP-GRAM-W and mCherry-D4H proteins was detected 4 hours after LDL treatment. The binding of both biosensors increased gradually over time during a 12-hour time course, supporting the occurrence of an increase in accessible cholesterol in both leaflets of the PM upon LDL treatment (Supplementary Fig. 3b–d). Incubation of cells with mCherry-D4H proteins for a longer duration resulted in more binding of mCherry-D4H proteins to the PM, which was accompanied by the dissociation of EGFP-GRAM-W from the PM (Supplementary Fig. 4a–c). This is consistent with the property of D4H to trap accessible cholesterol in the extracellular leaflet of the PM and reduce accessible cholesterol in the cytosolic leaflet of the PM[68,69]. Taken together, these results show that GRAM-W can be used as an effective biosensor to track the intracellular distribution of LDL-derived accessible cholesterol in live cells.

## GRAM-W detects reduction of accessible PM cholesterol mediated by rapamycin-dependent recruitment of GRAMD1b to ER-PM contacts

The levels of accessible cholesterol in the PM are constantly monitored by LTPs, including ER-anchored GRAMD1s, which can extract accessible cholesterol and transport it to the ER. To examine whether extraction of accessible PM cholesterol can be monitored by GRAM-W, GRAMD1b was acutely recruited to ER-PM contacts using rapamycin-induced dimerization of the FK506-binding protein (FKBP) and the FKBP-rapamycin-binding (FRB) domain[57,99]. HeLa cells stably expressing EGFP-GRAM-W were co-transfected with a PM-targeted FRB module (PM-FRB-mCherry) and with a chimeric GRAMD1b in which the GRAM domain was replaced by a miRFP-tagged FKBP

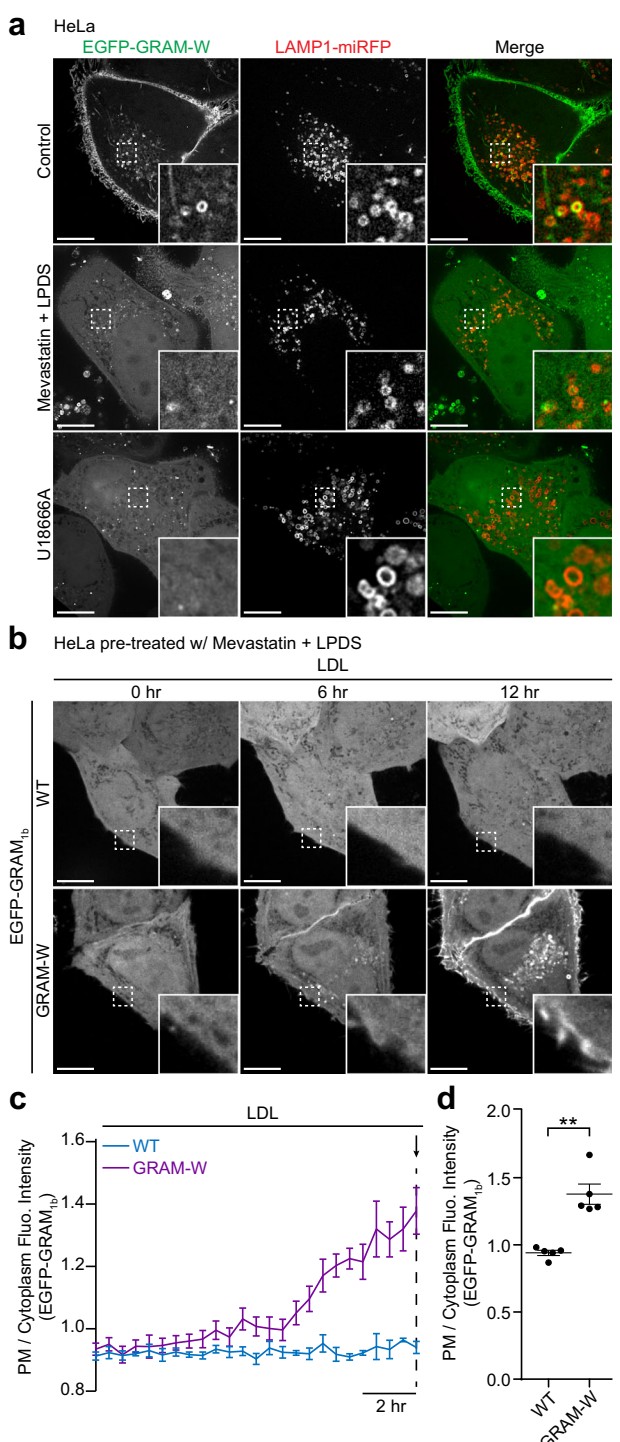

**Fig. 2 | GRAM-W detects LDL-derived accessible cholesterol in lysosomal membranes and the PM. a** Confocal images of live HeLa cells expressing EGFP-GRAM-W and LAMP1-miRFP (lysosomal marker) as indicated. Cells were incubated in either control medium, medium supplemented with mevastatin (50 μM) and 10% lipoprotein-deficient serum (LPDS), or medium containing NPC1 inhibitor (U18666A) (5 μM) for 16 hrs before imaging. Insets show at higher magnification the regions indicated by white dashed boxes. Note the decrease in colocalization between EGFP-GRAM-W and LAMP1-miRFP when cells are incubated in medium containing mevastatin and LPDS or medium containing U18666A. Scale bars, 10 μm. **b** Confocal images of live HeLa cells expressing either wild-type (WT) EGFP-GRAM$_{1b}$ or EGFP-GRAM-W that were stimulated by low-density lipoprotein (LDL) (50 μg/ml) over a time course of 12 hrs. Cells were pre-incubated in medium supplemented with mevastatin (50 μM) and 10% LPDS for 16 hrs to deplete cholesterol. Insets show at higher magnification the regions indicated by white dashed boxes. Note the gradual recruitment of EGFP-GRAM-W to the PM. Scale bars, 10 μm. **c** Time course of the ratio of PM signals to cytosolic signals of EGFP-GRAM$_{1b}$, as assessed by confocal microscopy and line scan analysis of HeLa cells expressing either wild-type EGFP-GRAM$_{1b}$ (WT) or EGFP-GRAM-W (GRAM-W), as shown in **b**. LDL treatment (50 μg/ml) is indicated. **d** Values of the ratio of PM signals to cytosolic signals of EGFP-GRAM$_{1b}$ at the end of the experiment as indicated by the arrow in **c** (mean ± SEM, $n = 5$ cells; data are pooled from one experiment for all conditions; two-tailed unpaired Student's $t$ test, **$P = 0.0005$).

used to monitor acute extraction of accessible cholesterol from the PM by LTPs.

### GRAM-W allows visualization of accessible PM cholesterol and reveals its source in multiple cell types

D4H is widely used as a cholesterol probe in live cell imaging. However, when EGFP-tagged D4H (EGFP-D4H) is expressed in the cytosol it binds only weakly to the PM at steady state in some cells[62,64–66] and forms numerous cytosolic aggregates when expressed in commonly used cell lines, including HeLa, U2OS (human osteosarcoma-derived cells), HEK293T (human embryonic kidney cells), and COS-7 (African green monkey kidney fibroblast-like cells) (Fig. 4a). Therefore, we examined whether GRAM-W can be used as an alternative to D4H in these different cell types.

We expressed EGFP-GRAM-W in U2OS cells (Fig. 4b), macrophages differentiated from THP-1 cells (human monocytes derived from an acute monocytic leukemia patient) (Fig. 4c), N/TERT cells (immortalized keratinocytes derived from fetal foreskin) (Fig. 4d), HEK293T (Fig. 4e), and COS-7 cells (Fig. 4f). In all these cell types, EGFP-GRAM-W bound to the PM at steady state. To determine the primary source of accessible PM cholesterol, as detected by EGFP-GRAM-W, in these cell types, cells expressing EGFP-GRAM-W were treated with mevastatin only, LPDS only, or the combination of mevastatin and LPDS. Mevastatin treatment did not reduce binding of EGFP-GRAM-W to the PM in all cell types tested, except for N/TERT cells (Fig. 4d). In N/TERT cells, mevastatin treatment resulted in near complete dissociation of EGFP-GRAM-W from the PM, suggesting that de novo cholesterol biosynthesis is the primary source of accessible PM cholesterol in N/TERT cells. By contrast, LPDS treatment caused major reductions in the binding of EGFP-GRAM-W to the PM in all cell types tested (again, except for N/TERT cells), suggesting that lipoproteins are the primary source of accessible PM cholesterol in U2OS, THP-1/macrophage, HEK293T, and COS-7 cells. In these cell types, binding of EGFP-GRAM-W to the PM was further reduced by combined treatment with LPDS and mevastatin, indicating that de novo cholesterol biosynthesis contributes to the maintenance of accessible PM cholesterol in the absence of lipoproteins in these cell types.

Collectively, these results show that GRAM-W allows visualization of accessible cholesterol in multiple cell types without forming cytoplasmic aggregates (as often observed with cytosolically expressed D4H). These imaging results are also in good agreement with the importance of de novo cholesterol biosynthesis in the maintenance of cellular cholesterol distribution in keratinocytes[100,101],

module (miRFP-FKBP-GRAMD1b), or miRFP-FKBP-GRAMD1b with a StART-like domain that cannot transport cholesterol [miRFP-FKBP-GRAMD1b (5P)] (Fig. 3a). TIRF microscopy revealed rapid recruitment of both miRFP-FKBP-GRAMD1b and miRFP-FKBP-GRAMD1b (5P) to the PM within 30 min of rapamycin treatment (Fig. 3b, c, Supplementary Fig. 5a, b, Supplementary Movie 2 and Supplementary Movie 3). The recruitment of miRFP-FKBP-GRAMD1b was followed by gradual dissociation of EGFP-GRAM-W from the PM (Fig. 3d, e, Supplementary Fig. 5a and Supplementary Movie 2). By contrast, EGFP-GRAM-W remained bound to the PM following the recruitment of miRFP-FKBP-GRAMD1b (5P) to the PM (Fig. 3d, e, Supplementary Fig. 5b and Supplementary Movie 3). These data demonstrate that GRAM-W can be

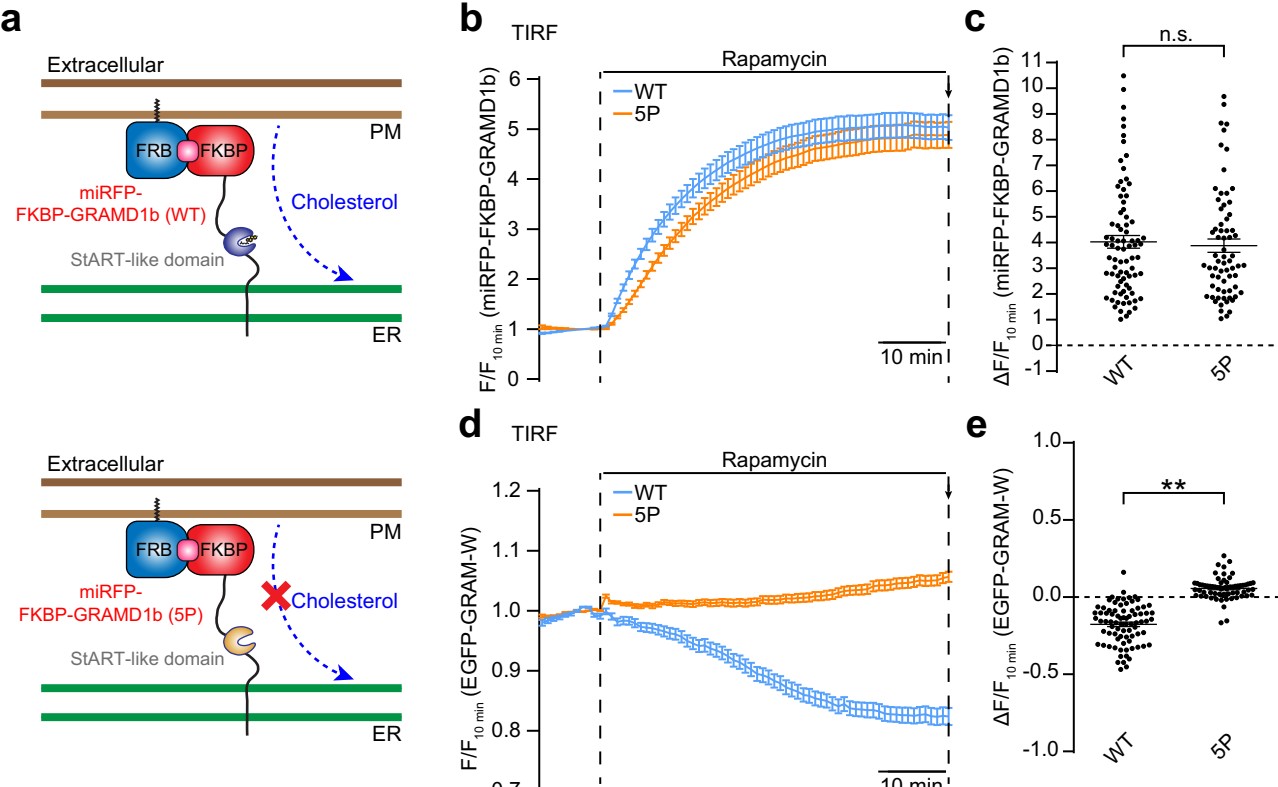

**Fig. 3 | GRAM-W detects reduction of accessible PM cholesterol mediated by rapamycin-dependent recruitment of GRAMD1b to ER-PM contacts.**
**a** Schematic representation of the rapamycin-induced GRAMD1b recruitment to the PM. GRAMD1b was rapidly recruited to the PM by rapamycin-induced dimerization of FRB and FKBP. A version of GRAMD1b with its GRAM domain replaced by a miRFP-tagged FKBP module [miRFP-FKBP-GRAMD1b (WT)] (top) or miRFP-FKBP-GRAMD1b (5P) (that is defective in cholesterol transport) (bottom) was expressed in HeLa cells expressing EGFP-GRAM-W together with mCherry-tagged FRB module that is targeted to the PM (PM-FRB-mCherry). **b** Time course of normalized miRFP signals in response to rapamycin, as assessed by TIRF microscopy of HeLa cells expressing EGFP-GRAM-W together with the indicated miRFP-FKBP-GRAMD1b constructs (WT or 5P) and PM-FRB-mCherry. Rapamycin addition

(200 nM) is indicated. **c** Values of $\Delta F/F_{10\,min}$ corresponding to the end of the experiment as indicated by the arrow in **b** [mean ± SEM, $n = 80$ cells (WT), $n = 68$ cells (5P); data are pooled from four independent experiments for WT and six independent experiments for 5P; two-tailed unpaired Student's $t$ test, n.s. denotes not significant]. **d** Time course of normalized EGFP signals in response to rapamycin, as assessed by TIRF microscopy of HeLa cells expressing EGFP-GRAM-W together with the indicated miRFP-FKBP-GRAMD1b constructs (WT or 5P) and PM-FRB-mCherry. Rapamycin addition (200 nM) is indicated. **e** Values of $\Delta F/F_{10\,min}$ corresponding to the end of the experiment as indicated by the arrow in **d** [mean ± SEM, $n = 80$ cells (WT), $n = 68$ cells (5P); data are pooled from four independent experiments for WT and six independent experiments for 5P; two-tailed unpaired Student's $t$ test, **$P < 0.0001$].

and with the fact that many other cell types rely heavily on lipoprotein-derived cholesterol for maintaining the distribution of cellular cholesterol[3,94,95].

## GRAM-W uncovers the contribution of OSBP to accessible cholesterol distribution in keratinocytes

Results from previous and our current studies suggest that keratinocytes heavily rely on de novo cholesterol biosynthesis for maintaining levels of accessible cholesterol in the PM. Thus, keratinocytes provide a unique system to study how cholesterol synthesized in the ER is delivered to the PM.

OSBP-mediated cholesterol transport at ER-Golgi contacts regulates the abundance of cholesterol in the Golgi and post-Golgi membranes, including the PM[28,102–105]. To investigate the potential role of OSBP in maintaining levels of accessible PM cholesterol in keratinocytes, N/TERT cells stably expressing EGFP-GRAM-W were treated with OSW-1 (a chemical inhibitor of OSBP) and the distribution of EGFP-GRAM-W was monitored using SDC microscopy. Gradual dissociation of EGFP-GRAM-W from the PM was observed over a 6-hour time course (Fig. 5a, b), supporting a critical role for OSBP in maintaining levels of accessible PM cholesterol. We confirmed this result using a cholesterol biosensor based on D4H. N/TERT cells treated with OSW-1 for different periods of time were incubated with purified

recombinant mCherry-D4H protein for 10 min, and its binding to the PM was analyzed by SDC microscopy. Binding of mCherry-D4H proteins to the PM was also reduced over a 6-hour time course of OSW-1 treatment, consistent with our measurements of accessible PM cholesterol via EGFP-GRAM-W (Fig. 5c, d).

Based on these results, we mutated the OSBP gene in N/TERT cells using CRISPR/Cas9-mediated gene editing. Immunoblotting showed a significant reduction in endogenous OSBP, confirming knock-out (KO) of OSBP in these cells (Supplementary Fig. 6a). OSBP KO N/TERT cells showed a major reduction in the binding of EGFP-GRAM-W to the PM, similar to OSW-1 treatment (Fig. 5e, g). Re-expression of mCherry-OSBP in these cells restored the localization of EGFP-GRAM-W to the PM (Fig. 5f, g), confirming the critical importance of OSBP in maintaining accessible PM cholesterol in keratinocytes. These results show that GRAM-W can be used to investigate the role of LTPs, including OSBP, in maintaining the distribution of cellular cholesterol.

## GRAM-W allows visualization of accessible cholesterol distribution in iPSC-derived neurons

Dysregulation of cholesterol distribution has been implicated in various neurodegenerative disorders, including Alzheimer's and Parkinson's diseases[2,4–7]. Recently, neurons derived from human iPSCs have been used as a model system to investigate the molecular

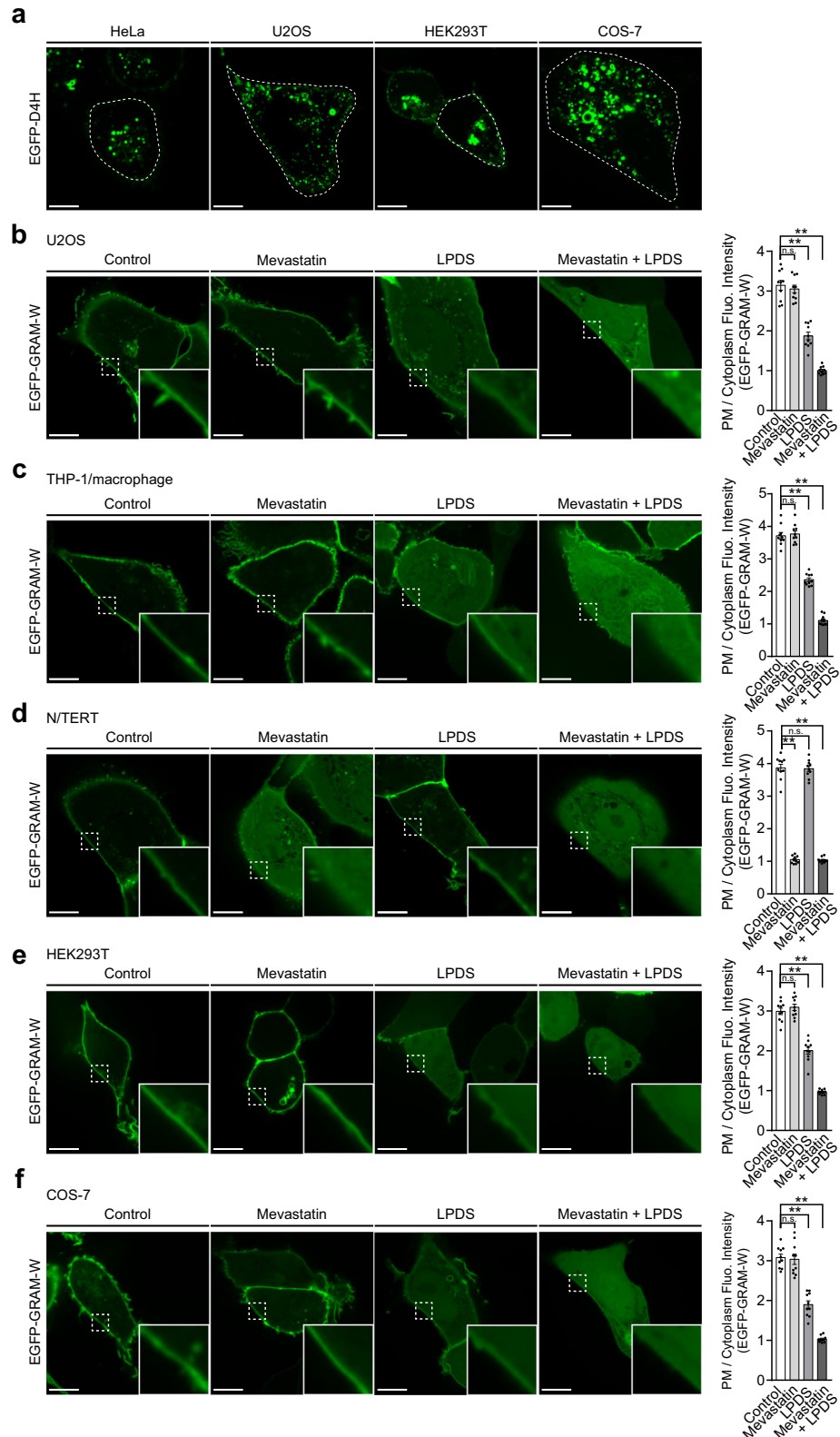

pathogenesis of various neurological disorders[106–110]. However, the distribution of accessible cholesterol in neurons remains poorly characterized due to limitations of existing tools. Thus, we asked whether our GRAM domain-based biosensors could be used in iPSC-derived neurons to visualize the distribution of accessible cholesterol.

We differentiated dopaminergic neurons, whose degeneration is implicated in Parkinson's disease, from human iPSCs and transduced them with lentiviruses at 28 days in vitro (DIV) to express WT EGFP-GRAM$_{1b}$, EGFP-GRAM-H, or EGFP-GRAM-W. Neurons were then imaged using SDC microscopy. Expression of EGFP-GRAM-W in dopaminergic neurons was confirmed by immunolabeling with antibodies against tyrosine hydroxylase (TH) (a marker of dopaminergic neurons)[111] and microtubule-associated protein 2 (MAP2) (a marker of mature neurons)[112]. EGFP-GRAM-W was expressed at high levels in

**Fig. 4 | GRAM-W allows visualization of accessible PM cholesterol and reveals its source in multiple cell types. a** Confocal images of indicated live cells transiently expressing EGFP-tagged D4H (EGFP-D4H). White dotted lines are drawn to depict the PM. Note the presence of EGFP-D4H aggregates in the cytoplasm of different cell types. Scale bars, 10 μm. **b**–**f** Confocal images of indicated live cells expressing EGFP-GRAM-W. EGFP-GRAM-W was either transiently expressed (U2OS, HEK293T, and COS-7) or stably expressed by lentiviral transduction (THP-1/macrophages and N/TERT). Cells were incubated in either control medium or medium supplemented with indicated components [mevastatin (50 μM); 10% lipoprotein-deficient serum (LPDS)] for 16 hrs before imaging. THP-1 cells were differentiated to macrophages by 3-day treatment with phorbol-12-myristate-13-acetate (PMA) (100 nM) prior to the experiment. Insets show at higher magnification the regions indicated by white dashed boxes. Scale bars, 10 μm. Right: Quantification of the ratio of PM signals to cytosolic signals of EGFP-GRAM-W, as assessed by confocal microscopy and line scan analysis (mean ± SEM, $n = 10$ cells for all conditions; data are pooled from one experiment; Dunnett's multiple comparisons test, **$P < 0.0001$. n.s. denotes not significant).

neurons that were positive for TH and MAP2 (Fig. 6a), showing that GRAM-W can indeed be expressed in iPSC-derived dopaminergic neurons.

Similar to results seen with other cell types, EGFP-GRAM-W expressed in iPSC-derived neurons bound more strongly to the PM than WT EGFP-GRAM$_{1b}$ or EGFP-GRAM-H (Fig. 6b, c). To validate that EGFP-GRAM-W senses accessible cholesterol in the neuronal PM, iPSC-derived neurons expressing EGFP-GRAM-W were treated with either mevastatin (to inhibit de novo cholesterol biosynthesis in the ER) or MCD (to deplete accessible PM cholesterol) (Fig. 6d, e). Mevastatin treatment did not affect binding of EGFP-GRAM-W to the PM. By contrast, MCD treatment significantly reduced binding of EGFP-GRAM-W to the PM (Fig. 6d, e). These results are consistent with previous studies reporting that mature neurons rely heavily on the uptake of lipoprotein-derived cholesterol rather than de novo cholesterol biosynthesis in the ER[113–116]. As the binding of EGFP-GRAM-W to the PM was reduced by MCD, our results show that EGFP-GRAM-W indeed can sense accessible cholesterol in the PM of neurons. Taken together, these analyses reveal that GRAM-W is a powerful tool for visualizing the distribution of accessible cholesterol in live iPSC-derived neurons.

## Ultrasensitive detection of accessible PM cholesterol via a ddFP-tagged GRAM domain

Dimerization-dependent fluorescent proteins (ddFPs) belong to a group of proteins comprised of two subunits that undergo reversible dimerization, leading to fluorescence only when they are dimerized. By targeting one protein subunit to the PM, proximity of the other subunit to the PM can be monitored via changes in fluorescence signals[76,77]. Unlike a typical fluorescent protein, such as EGFP or mCherry, ddFP allows elimination of cytosolic signals when one component is targeted to the PM. By combining this property of ddFPs with our GRAM domain-based biosensors, we generated a system that enables the ultrasensitive detection of the levels of accessible PM cholesterol via changes in fluorescence signals without the interference of cytosolic signals.

One ddFP subunit (RA) was targeted to the PM using the lipid modification motif of lymphocyte-specific protein tyrosine kinase (Lck). The other subunit (B) was fused with GRAM-W to detect accessible cholesterol (Fig. 7a). HeLa cells co-expressing PM-anchored RA and B-tagged GRAM-W (ddFP-R-GRAM-W system) exhibited red fluorescence specifically around the PM as monitored by SDC microscopy, indicating successful dimerization of the ddFP subunits at the PM (Fig. 7a, c). We also generated ddFP-R-GRAM-H system, in which B-tagged GRAM-H (which binds the PM at steady state, albeit less efficiently compared to GRAM-W) was co-expressed with PM-anchored RA (Fig. 7b). HeLa cells expressing the ddFP-R-GRAM-H system also exhibited red fluorescence around the PM, although ddFP-R-GRAM-H generally resulted in a weaker fluorescence signal compared to ddFP-R-GRAM-W (Fig. 7b, d). These results are consistent with the increased affinity of GRAM-W to accessible PM cholesterol compared to GRAM-H.

HeLa cells expressing ddFP-R-GRAM-W were treated with MCD to deplete accessible cholesterol from the PM. Within 5 min of MCD treatment, fluorescence signals associated with ddFP-R-GRAM-W were nearly eliminated (~10-fold reduction) (Fig. 7c, e, f and Supplementary Movie 4). On the other hand, HeLa cells expressing ddFP-R-GRAM-H were treated with the complex of MCD/Cholesterol to load cholesterol to the PM. Fluorescence signals associated with ddFP-R-GRAM-H dramatically increased within 10 min of MCD/Cholesterol treatment (~3.5 fold increase) (Fig. 7d, g, h and Supplementary Movie 5).

Finally, we asked if the ddFP-R-GRAM system could be coupled with flow cytometry to quantitatively measure levels of accessible PM cholesterol in a large population of live cells. HeLa cells expressing either ddFP-R-GRAM-W or ddFP-R-GRAM-H were treated with MCD, the combination of mevastatin and LPDS, or the complex of MCD/Cholesterol. ddFP-R-GRAM-W or ddFP-R-GRAM-H fluorescence was then measured using flow cytometry (Fig. 7i–n and Supplementary Fig. 7a, b). Compared to control non-treated HeLa cells, treatment with MCD for 15 min resulted in ~4 fold reduction in ddFP-R-GRAM-W fluorescence signals (Fig. 7i) and ~3 fold reduction in ddFP-R-GRAM-H fluorescence signals (Fig. 7l). Combined treatment with mevastatin and LPDS for 16 hours resulted in significant decrease (~2 fold) in both ddFP-R-GRAM-W and ddFP-R-GRAM-H fluorescence signals compared to control non-treated HeLa cells (Fig. 7j, m). In addition, HeLa cells expressing either ddFP-R-GRAM-W or ddFP-R-GRAM-H were treated with the complex of MCD/Cholesterol for 15 min, and ddFP-R-GRAM-W or ddFP-R-GRAM-H fluorescence was then measured. Treatment with MCD/Cholesterol slightly increased ddFP-R-GRAM-W signals (Fig. 7k), while it resulted in prominent increase (~2 fold) in ddFP-R-GRAM-H fluorescence signals compared to control non-treated HeLa cells (Fig. 7n).

These results demonstrate that the ddFP-R-GRAM system is a highly sensitive tool to measure levels of accessible PM cholesterol in live cells via simple detection of changes in fluorescence signals. They also show that the ddFP-R-GRAM-W system (and more generally GRAM-W) is most suitable for detecting the decrease in accessible cholesterol in the PM and that the ddFP-R-GRAM-H system (and more generally GRAM-H) is most suitable for detecting the increase of accessible cholesterol in the PM. Collectively, our GRAM domain-based biosensors serve as powerful tools for investigation of accessible cholesterol distribution in live cells.

## Discussion

It has been difficult to visualize accessible cholesterol within live cells due to the lack of an effective biosensor. In this study, we report a biosensor based on the GRAM domain of GRAMD1b (i.e., GRAM$_{1b}$) that is specific for accessible cholesterol. Our major findings are the following:

1. We performed a screen and identified a GRAM domain variant (GRAM-W) that strongly bound to the PM at steady state (even without additional loading of cholesterol). Using in vitro liposome binding assays, we found that purified GRAM-W proteins exhibited increased sensitivity for both accessible cholesterol and anionic lipids, such as PS.
2. Unlike EGFP-D4H, cytosolically expressed EGFP-GRAM-W did not form major aggregates. EGFP-GRAM-W detected accessible

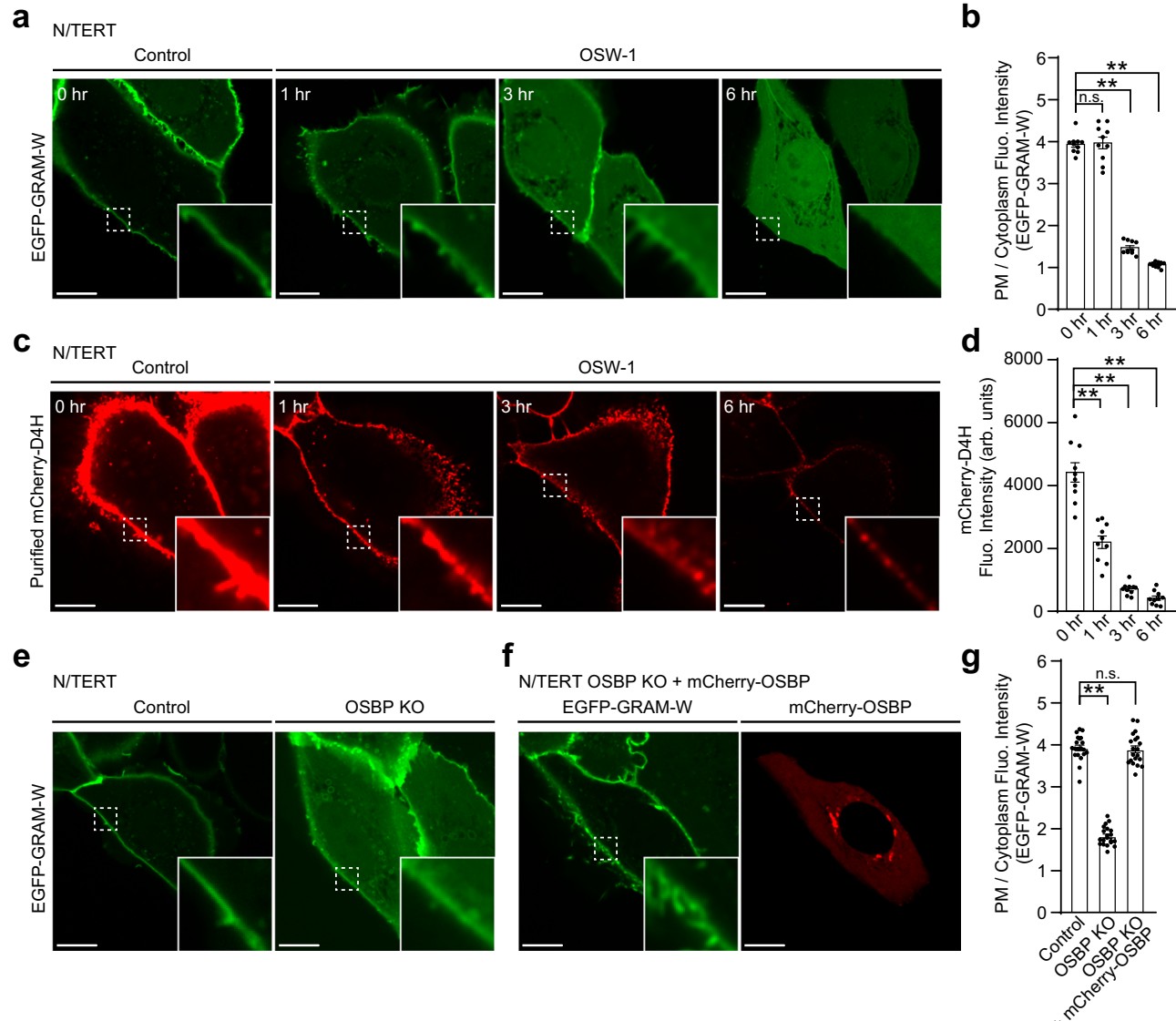

**Fig. 5 | Contribution of OSBP to accessible cholesterol distribution in keratinocytes as revealed by GRAM-W. a** Confocal images of live N/TERT cells stably expressing EGFP-GRAM-W. Cells were treated with OSW-1 (20 nM) over the time course of 6 hrs. Insets show at higher magnification the regions indicated by white dashed boxes. Scale bars, 10 μm. **b** Quantification of the ratio of PM signals to cytosolic signals of EGFP-GRAM-W, as assessed by confocal microscopy and line scan analysis as shown in **a** (mean ± SEM, $n = 10$ cells for all conditions; data are pooled from one experiment; Dunnett's multiple comparisons test, **$P < 0.0001$. n.s. denotes not significant). **c** Confocal images of live N/TERT cells that were treated with OSW-1 (20 nM) over the time course of 6 hrs. Cells were stained with purified mCherry-D4H proteins (15 μg/ml) for 10 min before imaging. Insets show at higher magnification the regions indicated by white dashed boxes. Scale bars, 10 μm. **d** Quantification of the mCherry-D4H signals, as assessed by confocal microscopy and line scan analysis as shown in **c** (mean ± SEM, $n = 10$ cells for all conditions; data are pooled from one experiment; Dunnett's multiple comparisons test, **$P < 0.0001$. **e** Confocal images of live wild-type N/TERT (Control) and OSBP knockout N/TERT (OSBP KO) cells stably expressing EGFP-GRAM-W. Insets show at higher magnification the regions indicated by white dashed boxes. Note the reduction of the binding of EGFP-GRAM-W to the PM in N/TERT OSBP KO cells. Scale bars, 10 μm. **f** Confocal images of live N/TERT OSBP KO cells stably expressing EGFP-GRAM-W together with mCherry-tagged OSBP (mCherry-OSBP). Insets show at higher magnification the regions indicated by white dashed boxes. Note that the expression of mCherry-OSBP restores the binding of EGFP-GRAM-W to the PM in N/TERT OSBP KO cells. Scale bars, 10 μm. **g** Quantification of the ratio of PM signals to cytosolic signals of EGFP-GRAM-W, as assessed by confocal microscopy and line scan analysis as shown in **e**, **f** (mean ± SEM, $n = 20$ cells for all conditions; data are pooled from two independent experiments; Dunnett's multiple comparisons test, **$P < 0.0001$. n.s. denotes not significant).

cholesterol in several cellular membranes, including the PM and lysosomal membranes. Importantly, binding of EGFP-GRAM-W to these membranes depended on lipoprotein-derived cholesterol and/or de novo synthesized cholesterol, showing that GRAM-W can be used to track accessible cholesterol in live cells.

3. EGFP-GRAM-W allowed visualization of accessible cholesterol distribution in multiple cell types, including iPSC-derived neurons. By monitoring the binding of EGFP-GRAM-W to the PM, it was possible to differentiate between cells that maintain

accessible PM cholesterol via lipoprotein-derived cholesterol and those that use de novo cholesterol synthesis. Using keratinocytes as a model, we showed that OSBP, a key LTP implicated in cholesterol transport, is critical for the transport of de novo synthesized cholesterol to the PM.

4. By combining the ddFP system with our GRAM domain-based biosensors for accessible cholesterol (GRAM-H and GRAM-W), we showed that the levels of accessible PM cholesterol can be selectively monitored in live cells via simple detection of

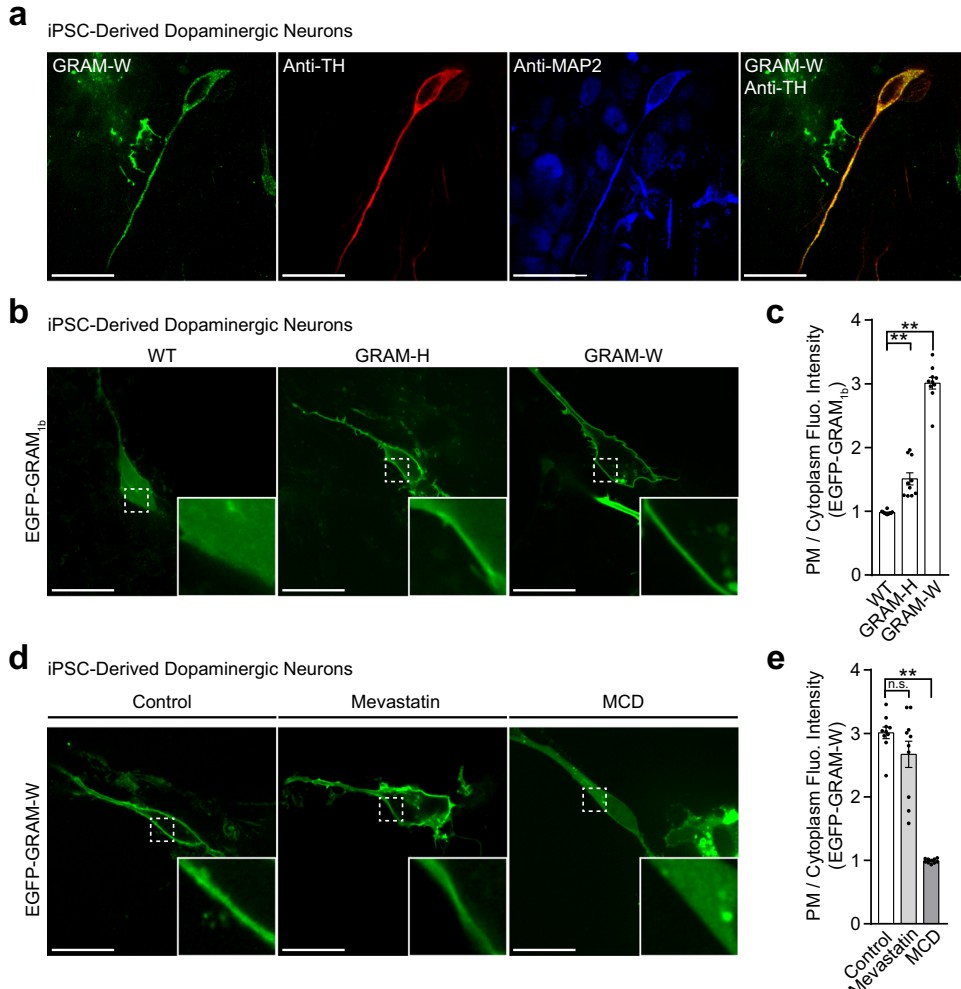

**Fig. 6 | GRAM-W allows visualization of accessible cholesterol distribution in iPSC-derived neurons. a** Confocal images of fixed human iPSC-derived dopaminergic neurons expressing EGFP-GRAM-W. Cells were immunolabeled with antibodies against tyrosine hydroxylase (Anti-TH) and microtubule-associated protein 2 (Anti-MAP2). Note the expression of EGFP-GRAM-W in dopaminergic neurons positive for TH and MAP2. Scale bars, 10 μm. **b** Confocal images of live human iPSC-derived neurons expressing either wild-type EGFP-GRAM$_{1b}$ (WT), EGFP-GRAM-H (GRAM-H), or EGFP-GRAM-W (GRAM-W) constructs as indicated. Insets show at higher magnification the regions indicated by white dashed boxes. Scale bars, 10 μm. **c** Quantification of the ratio of PM signals to cytosolic signals of EGFP-GRAM$_{1b}$, as assessed by confocal microscopy and line scan analysis as shown in

**b** (mean ± SEM, $n = 10$ cells for all conditions; data are pooled from one experiment; Dunnett's multiple comparisons test, **$P < 0.0001$). **d** Confocal images of live human iPSC-derived neurons expressing EGFP-GRAM-W that were incubated with either control medium or medium supplemented with mevastatin (50 μM) for 16 hrs or MCD (5 mM) for 20 min before imaging. Insets show at higher magnification the regions indicated by white dashed boxes. Scale bars, 10 μm. **e** Quantification of the ratio of PM signals to cytosolic signals of EGFP-GRAM-W, as assessed by confocal microscopy and line scan analysis as shown in **d** (mean ± SEM, $n = 10$ cells for all conditions; data are pooled from one experiment; Dunnett's multiple comparisons test, **$P < 0.0001$. n.s. denotes not significant).

changes in fluorescence signals. ddFP is reversible, allowing us to detect both increases and decreases of accessible PM cholesterol.

The mechanisms by which cells maintain the distribution of cholesterol are being studied intensely. Thus, a number of tools have been developed to visualize and track cellular cholesterol. Apart from filipin and CDCs, such as PFO and ALO, dehydroergosterol (DHE), a naturally occurring fluorescent cholesterol analog, and cholesterol molecules conjugated with a fluorophore, such as NBD and TopFluor, have been commonly used to study the dynamics of intracellular cholesterol distribution and transport. However, DHE, which is a derivative of ergosterol, is not structurally identical to cholesterol, and therefore, it may not precisely reflect the distribution of endogenous cholesterol[55,117,118]. Attachment of a large fluorophore, such as NBD and TopFluor, to cholesterol can modify the physical and biochemical properties of cholesterol, likely altering its behavior and distribution within cellular membranes[55,119,120]. A genetically encoded cholesterol

biosensor, such as PFO-D4, overcomes these limitations via direct detection of endogenous cholesterol molecules. While PFO-D4 has gained popularity as an alternative tool to visualize the distribution of accessible cholesterol in cells, one major limitation of PFO-D4 is that cytosolically expressed PFO-D4 binds only weakly to the PM, where cholesterol is highly enriched[63–66]. It has been proposed that the inefficient binding of PFO-D4 to the cytosolic leaflet of the PM is caused by electronic repulsion between the negatively charged surface of the PFO-D4 and the anionic head groups of phospholipids, such as PS, that are enriched in the cytosolic leaflet of the PM[62]. Mutation of an aspartic acid residue, in particular D434, to serine or alanine enhances the binding of cytosolically expressed PFO-D4 to the PM, supporting this notion[60,63,121,122]. However, a large fraction of EGFP-D4H is still present in the cytosol, where it spontaneously forms aggregates (Fig. 4a). This makes it difficult to use EGFP-D4H for determining the subcellular distribution of accessible cholesterol in live cells.

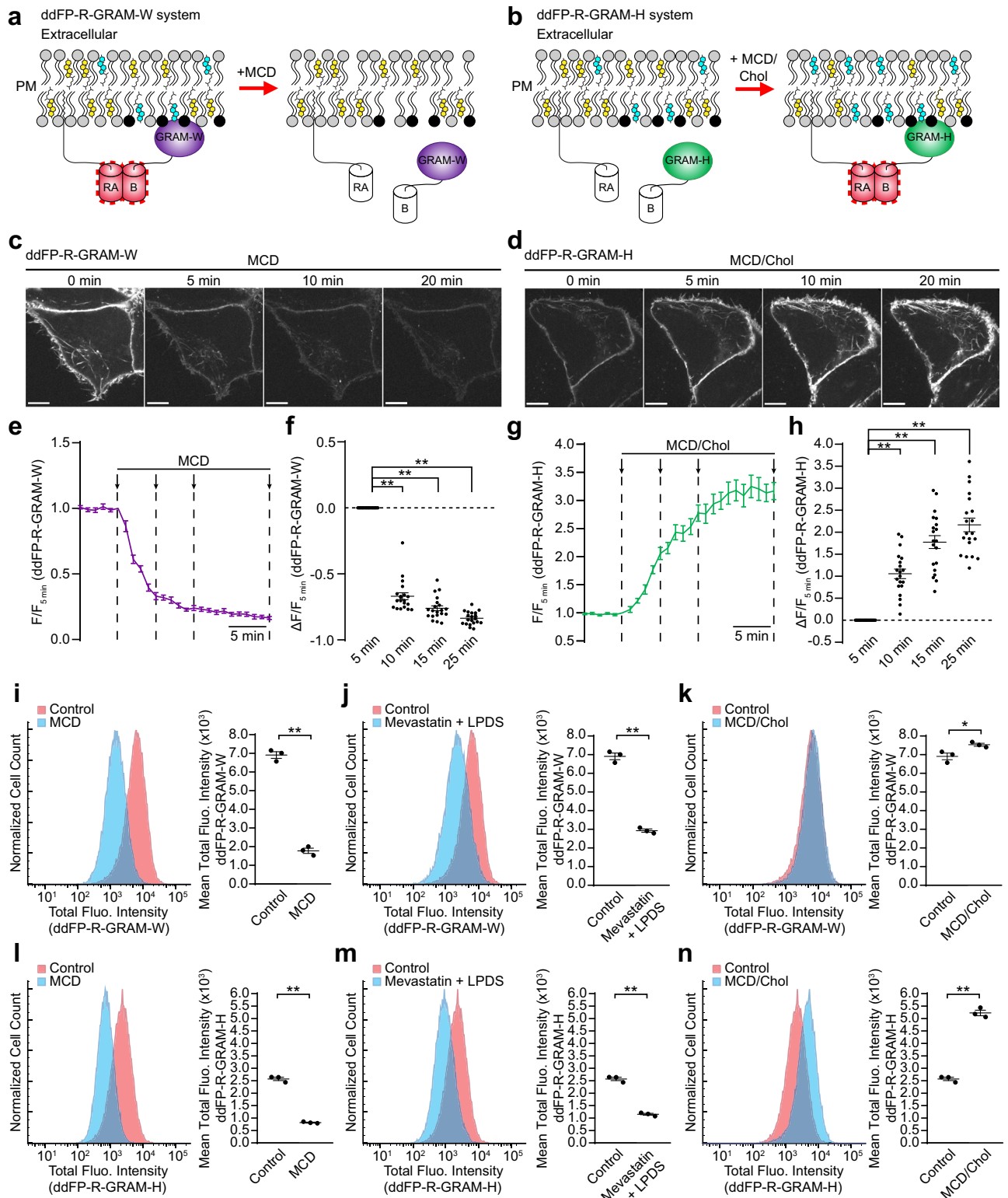

By contrast, GRAM$_{1b}$ is a co-incidence detector of accessible cholesterol and anionic lipids, and therefore binds most efficiently to membranes that contain both cholesterol and anionic lipids, such as the PM. In our previous study, we characterized the biochemical properties of GRAM$_{1b}$ and found that the glycine residue at position 187 (G187) plays a key role in determining the sensitivity of this domain to accessible cholesterol[68]. In the current study, we found that GRAM-W (i.e., GRAM$_{1b}$ carrying the G187W mutation) exhibits enhanced binding to the PM compared to WT GRAM$_{1b}$ or GRAM-H (GRAM$_{1b}$ carrying the G187L mutation). Interestingly, purified GRAM-W protein exhibited increased sensitivity to both cholesterol and PS, unlike the GRAM-H protein, which is more sensitive to accessible cholesterol (compared with WT GRAM$_{1b}$) but its sensitivity to PS is unaltered[68]. Hence, the G187W mutation dramatically enhanced the ability of GRAM$_{1b}$ to detect accessible cholesterol in cellular membranes that contain large amounts of PS, such as the PM. Further, GRAM-W did not

**Fig. 7 | Ultrasensitive detection of accessible PM cholesterol via a ddFP-tagged GRAM domain. a, b** Schematic representation of accessible PM cholesterol detection using GRAM-W or GRAM-H and a red dimerization-dependent fluorescent protein (ddFP-R). PM-anchored RA (PM-RA) and B-fused with either GRAM-W (B-GRAM-W) or GRAM-H (B-GRAM-H) were co-expressed in HeLa cells. Dimerization of "PM-RA" with "B-GRAM-W" or "B-GRAM-H" results in red fluorescence. **c, d** Confocal images of live HeLa cells expressing either ddFP-R-GRAM-W system that were treated with MCD (5 mM) (**c**) or ddFP-R-GRAM-H system that were treated with MCD/Chol (200 μM) (**d**) over a time course of 20 min. Scale bars, 10 μm. **e** Time course of normalized fluorescence intensity of ddFP-R-GRAM-W, as assessed by confocal microscopy as shown in **c. f** Values of $\Delta F/F_{5\,min}$ corresponding to different time points of the experiment as indicated by the arrows in **e** (mean ± SEM, $n = 20$ cells from each indicated time point; data are pooled from four independent experiments; Dunnett's multiple comparisons test, **$P < 0.0001$. **g** Time course of normalized fluorescence intensity of ddFP-R-GRAM-H, as assessed by confocal microscopy as shown in **d. h** Values of $\Delta F/F_{5\,min}$ corresponding to different time points of the experiment as indicated by the arrows in **g** (mean ± SEM, $n = 20$ cells from each indicated time point; data are pooled from four independent experiments; Dunnett's multiple comparisons test, **$P < 0.0001$. **i–n** Profiling of ddFP-R-GRAM-W fluorescence intensity (**i–k**) or ddFP-R-GRAM-H fluorescence intensity (**l–n**) in HeLa cells stably expressing either ddFP-R-GRAM-W system or ddFP-R-GRAM-H system. Cells were either incubated with control medium or medium supplemented with MCD (5 mM) for 15 min (**i, l**), incubated with control medium or medium supplemented with mevastatin (50 μM) and 10% lipoprotein-deficient serum (LPDS) for 16 hrs (**j, m**), or incubated with control medium or medium supplemented with MCD/Chol (200 μM) for 15 min (**k, n**), and assessed by flow cytometry. Right: Values of the mean total fluorescence intensity of HeLa (mean ± SEM, $n = 3$ independent experiments for all conditions; two-tailed unpaired Student's $t$ test, **$P < 0.0001$, *$P = 0.032$). The same set of controls was used between **i–k** and between **l–n**.

form cytoplasmic aggregates, overcoming a limitation of PFO-D4H. Although GRAM-W is a promising tool for studying the distribution of accessible PM cholesterol, its strong binding to the PM at steady state makes it potentially difficult to detect further increases in the levels of accessible cholesterol in the PM. This limitation can be overcome via GRAM-H, which binds more strongly to the PM only when levels of accessible cholesterol are elevated in this bilayer (see Fig. 7). While the expression of GRAM-W in our assays did not significantly alter the movement of accessible cholesterol in cells and the organization of accessible cholesterol in the PM, it might potentially affect other aspects of cholesterol membrane biology. Importantly, the binding of GRAM domain-based accessible cholesterol biosensors to some organelles with low levels of anionic lipids in cytosolic leaflets, such as the ER, are likely inefficient even in the presence of accessible cholesterol. Existing probes, including GRAM-W, GRAM-H, and PFO-D4H, are complementary, and we recommend using several approaches for monitoring intracellular cholesterol distribution in live cells.

Using GRAM-W, we successfully visualized accessible cholesterol in the PM and in lysosomal membranes. In addition, we showed the critical importance of NPC1, whose dysfunction is associated with neurodegeneration, in maintaining accessible pool of cholesterol in lysosomal membranes for the maintenance of intracellular cholesterol distribution. While GRAM-W did not bind to other organelles at steady state, such as the *trans*-Golgi network (TGN) and mitochondria, GRAM-W may detect accessible cholesterol in these organelles when levels of cholesterol are acutely or chronically elevated. Further, GRAM-W detected accessible cholesterol in a number of different cell types, including U2OS cells, THP-1/macrophages, N/TERT cells, and iPSC-derived dopaminergic neurons. In N/TERT cells, inhibition of de novo cholesterol biosynthesis by mevastatin significantly reduced the binding of EGFP-GRAM-W to the PM, suggesting that de novo cholesterol biosynthesis is the main source of accessible PM cholesterol in keratinocytes. We further showed that N/TERT cells use OSBP, a major LTP involved in PI4P-driven counter transport of cholesterol at membrane contact sites, to maintain levels of accessible cholesterol in the PM, providing mechanistic insights into the process of cholesterol transport in keratinocytes. By contrast, binding of EGFP-GRAM-W to the PM in iPSC-derived dopaminergic neurons (and other cell types tested in this study) was not reduced by mevastatin. This is consistent with previous studies showing that mature neurons stop synthesizing cholesterol and instead depend on an external supply of cholesterol, such as lipoproteins derived from glial cells[113–116]. Collectively, we show that GRAM-W is a valuable tool for visualizing accessible cholesterol in live neurons. Such a tool will allow us to investigate how neurons maintain intracellular cholesterol distribution and examine the dysregulation of this process (as well as cholesterol metabolism) in a range of neuronal pathologies.

Finally, we combined ddFP and the GRAM domain variants, GRAM-W and GRAM-H, to develop an ultrasensitive method for measuring levels of accessible PM cholesterol by simply monitoring changes in ddFP fluorescence. Importantly, this method allowed us to quantitatively measure levels of accessible PM cholesterol in large population of live cells via flow cytometry. ddFP has been successfully used to monitor dynamic changes in the levels of other membrane lipids, such as PI(4,5)P$_2$, in the PM[76,77]. Here, we combined ddFPs with GRAM domains to achieve selective detection of the levels of accessible PM cholesterol via changes in total fluorescence. While our current system monitors accessible cholesterol specifically at the PM, it should be possible to target one component of ddFP (RA) to other organelle membrane, such as lysosomal or TGN membranes, to measure levels of accessible cholesterol in other organelles in live cells.

In conclusion, we have developed novel genetically encoded biosensors for accessible cholesterol by engineering GRAM$_{1b}$. The distribution of cellular cholesterol is tightly controlled via the regulated transport of accessible cholesterol between membranes, and alterations in the distribution of cholesterol can significantly impact numerous cellular processes, including signal transduction, membrane trafficking, and lipid metabolism. Our GRAM domain-based accessible cholesterol biosensors will enable the field to gain important insights into the molecular mechanisms of cholesterol transport, paving the way toward uncovering the pathogenesis of diseases related to dysregulation of cholesterol homeostasis.

## Methods

### Antibodies and chemicals
Primary and secondary antibodies, chemicals, lipids, and other reagents used in this study are listed in Supplementary Table 1.

### DNA plasmids
DNA plasmids, the sequences of oligos and primers used are listed in Supplementary Table 1.

**For recombinant protein purification**
**Cloning of His-GRAM$_{1b}$, His-GRAM$_{1b}$ (G187L), and His-GRAM$_{1b}$ (G187W).** The His-GRAM$_{1b}$ and His-GRAM$_{1b}$ G187L plasmids were previously generated[68]. The amino acid residue G187 present in the GRAM$_{1b}$ was mutated using site-directed mutagenesis in pNIC28-Bsa4 His-GRAMD1b$_{92-207}$ (GRAM$_{1b}$) with a primer set, GRAMD 1b_G187W_F and GRAMD1b_G187W_R, to generate pNIC28-Bsa4 His-GRAM1b (G187W).

**Cloning of mCherry-D4H.** The mCherry-D4H plasmid was previously generated[68].

**For mammalian expression**
**Cloning of EGFP-GRAM$_{1b}$ G187 variants.** The amino acid residue (G187) present in the GRAM$_{1b}$ was previously systematically mutated to

every other amino acid using site-directed mutagenesis in EGFP-GRAM$_{1b}$[68].

**Cloning of PM-FRB-mCherry, miRFP-FKBP-GRAMD1b (WT), miRFP-FKBP-GRAMD1b (5P), and EGFP-D4H.** The PM-FRB-mCherry plasmid was previously generated[57]. The miRFP-FKBP-GRAMD1b (WT) and (5P) plasmids were previously generated[57]. The EGFP-D4H plasmid was previously generated[28].

**Cloning of PM-RA.** cDNA of RA of RA-NES (Addgene #61019) was amplified by PCR using a primer set, AgeI-RA-S and XhoI-RA-AS. The PCR product was digested and ligated into Lck-mScarlet-I (Addgene #98821) at AgeI and XhoI sites to generate PM-RA.

**Cloning of B-GRAM-W and B-GRAM-H.** cDNA of B-GRAM-W (IDT gBlock: EcoRI_GB-GRAM-W_BamHI) and that of B-GRAM-H (IDT gBlock: EcoRI_GB-GRAM-H_BamHI) were amplified by PCR using a primer set, AgeI-GB3-NS and EcoRI-stop-GRAM1b-AS. The PCR products were digested and ligated into pEGFP-C1 at AgeI and BamHI sites to generate B-GRAM-W and B-GRAM-H, respectively.

**For lentivirus generation**

**Cloning of pLJM1-EGFP-GRAM$_{1b}$, pLJM1-EGFP-GRAM-H and pLJM1-EGFP-GRAM-W.** EGFP-GRAM$_{1b}$, EGFP-GRAM$_{1b}$ (G187L) and EGFP-GRAM$_{1b}$ (G187W) were individually digested using NheI and MfeI and ligated into pLJM1-EGFP at NheI and EcoRI sites to generate pLJM1-EGFP-GRAM$_{1b}$, pLJM1-EGFP-GRAM-H, and pLJM1-EGFP-GRAM-W, respectively.

**Cloning of pLJM1-mCherry-OSBP.** cDNA of OSBP (NP_002547) from mRuby-OSBP[57] was digested and ligated at HindIII and BamHI sites in the mCherry-C1. Nucleotides encoding amino acid residues corresponding to S416 and L444 was then mutated using site-directed mutagenesis with the following primer sets, OSBP_S416S_F, OSBP_S416S_R, OSBP_L444L_F, and OSBP_L444L_R, to generate a variant of mCherry-OSBP that is resistant to CRISPR/Cas9 activity. The mCherry-OSBP was then digested using NheI and MfeI and ligated into pLJM1-EGFP at NheI and EcoRI sites to generate pLJM1-mCherry-OSBP.

**Cloning of pLJM1-PM-RA-BlastR.** cDNA of Blasticidin resistant gene of pLJC6-3HXA-TMEM192 (Addgene #104434) was PCR amplified using a primer set, BamHI-BlastR-NS and KpnI-BlastR-CAS. The PCR product was digested and ligated into pLJM1-EGFP at BamHI and KpnI sites to generate pLJM1-EGFP-BlastR. Subsequently, PM-RA was digested from PM-RA vector using NheI and EcoRI and ligated into pLJM1-EGFP-BlastR at NheI and EcoRI sites to generate pLJM1-PM-RA-BlastR.

**Cloning of pLJM1-B-GRAM-W-NeoR and pLJM1-B-GRAM-H-NeoR.** cDNA of Neomycin resistant gene of pEGFP-C1 was PCR amplified using a primer set, BamHI-NeoR-NS and KpnI-NeoR-CAS. The PCR product was digested and ligated into pLJM1-EGFP at BamHI and KpnI sites to generate pLJM1-EGFP-NeoR. Subsequently, B-GRAM-W from B-GRAM-W vector and B-GRAM-H from B-GRAM-H vector were digested using NheI and EcoRI and ligated into pLJM1-EGFP-NeoR at NheI and EcoRI sites to generate pLJM1-B-GRAM-W-NeoR and pLJM1-B-GRAM-H-NeoR, respectively.

## Cell culture and transfection

HeLa (Gift from Pietro De Camilli), U2OS (Gift from Wenting Zhao), HEK293T (Gift from Nguan Soon Tan), and COS-7 (Gift from Pietro De Camilli) cells were cultured in Dulbecco's modified Eagle's medium (DMEM) containing 20% fetal bovine serum (FBS) and 1% penicillin/streptomycin at 37 °C and 5% CO$_2$. THP-1 (Gift from Franklin L. Zhong) cells were cultured in Roswell Park Memorial Institute (RPMI) containing 10% FBS and 1% penicillin/streptomycin at 37 °C

and 5% CO$_2$. Phorbol 12-myristate-13-acetate (PMA) (Sigma-Aldrich/Merck) (100 nM) was added to THP-1 cells for 3 days to differentiate THP-1 into macrophages for confocal microscopy imaging. Keratinocytes (N/TERT cells) (Gift from James G. Rheinwald) were cultured in Keratinocyte serum-free medium (KSFM) supplemented with human recombinant epidermal growth factor (rEGF), bovine pituitary extract (BPE) as prepared according to manufacturer protocol, and 1% penicillin/streptomycin at 37 °C and 5% CO$_2$. N/TERT-1 cells are a kind gift from James G. Rheinwald and obtained under Material Transfer Agreement (Zhong lab, SRIS, Singapore). Transfection of plasmids was carried out with Lipofectamine 2000 (Thermo Fisher Scientific).

iPSCs from the asF5 line (CellResearch Corporation Pte Ltd) were maintained at 5% CO$_2$ and 37 °C in TeSR™-E8™ media (Stemcell Technologies) on Matrigel™ (Corning)-coated 6 well plates. Medium was changed daily, and cells were passaged upon reaching 70% confluency. For passaging, iPSCs were washed once with PBS and incubated with ReLeSR™ (Stemcell Technologies) for 5 min at 37 °C. Once colonies started to dissociate in clumps, TeSR™-E8™ media supplemented with 10 µM ROCK Inhibitor Y-27632 was added. 900 ml of TeSR™-E8™ media with 10 µM ROCK Inhibitor Y-27632 was added for each 100 ml of ReLeSR™ used. The contents were transferred to new Matrigel™-coated 6 well plates at a routine split ratio of 1:10. The wells were replaced with fresh TeSR™-E8™ media without Y-27632 approximately 24 hours later.

All cell lines were routinely verified as free of mycoplasma contamination at least every two months, using MycoGuard Mycoplasma PCR Detection Kit (Genecopoeia). No cell lines used in this study were found in the database of commonly misidentified cell lines that is maintained by ICLAC and NCBI Biosample.

## Generation of iPSC-derived dopaminergic neurons

iPSCs were washed with PBS and dissociated with StemPro™ Accutase™ (Gibco) by incubation at 37 °C for 5 to 8 min. To quench the Accutase™ reaction, 4 ml of TeSR™-E8™ media (Stemcell Technologies) with 10 µM ROCK Inhibitor Y-27632 was added for each 1 ml of Accutase™ used. The contents were collected into a 15 ml conical tube and centrifuged at 200 g for 5 min. The cell suspension was aspirated, and the cell pellet was resuspended in 1 ml TeSR™-E8™ media with 10 µM Y-27632 for cell counting. Thereafter, 1 million iPSCs were seeded in each well of a six-well plate coated with Matrigel™ and maintained in TeSR™-E8™ supplemented with Y-27632 for 24 hrs.

On initiation day 0, iPSC media was aspirated, and cells were cultured in SRM medium (KnockOut™-DMEM/F12, 15% KnockOut™ Serum Replacement, 1% GlutaMAX™, 1% MEM-Non-Essential Amino Acids Solution, 1% Sodium Bicarbonate 7.5% Solution, 0.1% 2-Mercaptoethanol) (Gibco) containing 100 nM LDN193189 and 10 µM SB431542.

Daily changes of culture media were performed according to the following media composition: on days 1 and 2, cells were cultured in SRM media with 100 nM LDN193189, 10 µM SB431542, 100 ng/ml recombinant mouse sonic hedgehog/Shh (C25II), and 2 µM Purmorphamine; on days 3 and 4, SRM media with 100 nM LDN193189, 10 µM SB431542, 100 ng/ml recombinant mouse sonic hedgehog/Shh (C25II), 2 µM Purmorphamine, and 3 µM CHIR99021; on days 5 and 6, 75% SRM media with 25% N2 media (Neurobasal™ media, 1% N-2 Supplement, 2% B-27™ Supplement without vitamin A, 1% GlutaMAX™) (Gibco) with 100 nM LDN193189, 100 ng/ml recombinant mouse sonic hedgehog/Shh (C25II), 2 µM Purmorphamine, and 3 µM CHIR99021; on days 7 and 8, 50% SRM with 50% N2 media supplemented with 100 nM LDN193189 and 3 µM CHIR99021; on days 9 and 10, 25% SRM with 75% N2 media supplemented with 100 nM LDN193189 and 3 µM CHIR99021; on days 11 and 12, NB/B27 media (Neurobasal™ media, 2% B-27™ Supplement without vitamin A, 1% GlutaMAX™) (Gibco) supplemented with 3 µM CHIR99021, 20 ng/ml BDNF, 20 ng/ml GDNF,

0.5 mM dibutyryl cAMP, 0.2 mM ascorbic acid, 1 ng/ml TGF-β3, and 10 nM DAPT.

On day 13, cells were dissociated with StemPro™ Accutase™ and passaged onto plates coated with 15 µg/ml poly-l-ornithine, 1 µg/ml fibronectin and 2 µg/ml laminin. Cultures were maintained in NB/B27 media with 20 ng/ml BDNF, 20 ng/ml GDNF, 0.5 mM dibutyryl cAMP, 0.2 mM ascorbic acid, 1 ng/ml TGF-β3, 10 nM DAPT, and 10 µM Y-27632 for 48 hours before culturing the cells in the same media composition without Y-27632 from day 15 onwards.

Cultures were passaged again on day 20 to make space once cells became bipolar. About 300,000 cells were plated per well on a dry 24 well plate that was coated with 15 µg/ml poly-l-ornithine, 1 µg/ml fibronectin and 2 µg/ml laminin. Cultures were fed every other day with the same day 15 media composition until desired end point.

### Generation of cell lines stably expressing either EGFP-GRAM$_{1b}$, EGFP-GRAM-H, EGFP-GRAM-W, mCherry-OSBP, or PM-RA & B-GRAM-W or PM-RA & B-GRAM-H

Lentiviral helper plasmids (pMD2.G, pRSV-REV, and pMDL/pRRE) (3.7 µg each) were transfected together with either one of the following plasmids (pLJM1-EGFP-GRAM$_{1b}$, pLJM1-EGFP-GRAM-H, pLJM1-EGFP-GRAM-W, pLJM1-mCherry-OSBP, pLJM1-PM-RA-BlastR, pLJM1-B-GRAM-W-NeoR, and pLJM1-B-GRAM-H-NeoR) (7.4 µg) into 4.4 × 10⁶ HEK293T cells according to manufacturer's protocol. Supernatant was collected 48 hrs after transfection and filtered using a 0.45 µm filter unit to recover lentiviruses. Filtered lentiviruses were then centrifuged for 16 hrs at 16,000 g and the pellet was resuspended in 1 mL of cell culture medium. Virus titer was measured using Lenti-X™ GoStix™ (Takara Bio) according to manufacturer's protocol. Approximately 100 µl of 1967 ng/ml p24 [GoStix Value (GV)] equivalent virus was used to transduce cells for 48 hrs.

To generate HeLa cells stably expressing either EGFP-GRAM$_{1b}$ or EGFP-GRAM-W, wild-type control HeLa cells were seeded at 1.5 × 10⁵ cells and transduced with lentiviruses (either LV-EGFP-GRAM$_{1b}$ or LV-EGFP-GRAM-W) to generate stable cell lines expressing indicated proteins. After puromycin selection (5 µg/ml), BD FACSAria™ Fusion (BD Biosciences) was used to isolate cells that are positive for EGFP fluorescence. The sorted cells were subsequently maintained in 0.5 µg/ml puromycin, and protein expression was confirmed by microscopy.

To generate THP-1 stable cell line expressing pLJM1-EGFP-GRAM-W, wild-type control THP-1 were seeded at 1.5 × 10⁵ cells and transduced with lentivirus (LV-EGFP-GRAM-W). BD FACSAria™ Fusion (BD Biosciences) was used to isolate cells that are positive for EGFP fluorescence. Protein expression in the sorted cells was then confirmed by microscopy.

To generate N/TERT cells stably expressing EGFP-GRAM-W, wild-type control N/TERT cells were seeded at 1.5 × 10⁵ cells and transduced with lentivirus (LV-EGFP-GRAM-W). After puromycin selection (5 µg/ml), cells were maintained in 0.5 µg/ml puromycin, and protein expression was confirmed by microscopy.

To generate OSBP knockout (KO) N/TERT cells stably expressing mCherry-OSBP, OSBP KO N/TERT cells stably expressing EGFP-GRAM-W were seeded at 1.5 × 10⁵ cells and transduced with lentivirus (LV-mCherry-OSBP). After puromycin selection (5 µg/ml), cells were maintained in 0.5 µg/ml puromycin, and protein expression for both EGFP-GRAM-W and mCherry-OSBP was confirmed by microscopy.

For the experiments with iPSC-derived dopaminergic neurons, day in vitro (DIV) 28 iPSC-derived dopaminergic neurons were transduced with lentiviruses (LV-EGFP-GRAM$_{1b}$, LV-EGFP-GRAM-H, or LV-EGFP-GRAM-W) to express indicated proteins. Protein expression for EGFP-GRAM$_{1b}$, EGFP-GRAM-H, or EGFP-GRAM-W in iPSC-derived dopaminergic neurons was confirmed by microscopy at DIV 30.

To generate HeLa cells stably expressing PM-RA and either B-GRAM-W or B-GRAM-H, wild-type control HeLa cells were seeded at

1.5 × 10⁵ cells and transduced with lentiviruses (LV-PM-RA and either LV-B-GRAM-W or LV-B-GRAM-H) to generate stable cell lines expressing indicated proteins. Cells positive for ddFP-R fluorescence were selected using G418 (800 µg/ml) and blasticidin (1 µg/ml), and BD FACSAria™ Fusion (BD Biosciences) was used to isolate cells that are positive for fluorescence. The sorted cells were subsequently maintained in 80 µg/ml G418, and protein expression was confirmed by microscopy.

### Generation of CRISPR/Cas9-edited cell lines

OSBP KO N/TERT cells were generated using the Alt-R CRISPR-Cas9 System (Integrated DNA Technologies, Inc.) and electroporation by the nucleofector system (Lonza) according to manufacturer's protocol. In brief, CRISPR gRNAs targeting OSBP genomic sequences AATG ACTTGATAGCTAAGCA (OSBP-sgRNA#1) and GCTCGAGGGTTTCTTCC AGT (OSBP-sgRNA#2) based on ref. 123 were designed (refer to Supplementary Table 1 for CRISPR-Cas9 sgRNA sequence). CRISPR-Cas9 ribonucleoprotein (RNP) complex consisting of the sgRNA (OSBP-sgRNA#1 and OSBP-sgRNA#2), and Cas9 nuclease (Integrated DNA Technologies, Inc.) was added to 6.5 × 10⁵ N/TERT cells suspended in P3 Primary Cell Nucleofector Solution and supplement (Lonza). Electroporation was carried out using the program "DS-138". Electroporated N/TERT cells suspension was then plated and efficiency of OSBP KO was confirmed by immunoblotting after 2 passages.

### Fluorescence microscopy

For imaging experiments, cells were plated onto 35 mm glass bottom dishes at low density (MatTek Corporation). For transiently expressed proteins in transfected cell lines, live-cell imaging was carried out one day after transfection unless otherwise stated.

Spinning disc confocal (SDC) microscopy and SDC-structured illumination microscopy (SDC-SIM) (Figs. 1b, c, g, h, 2a–d, 4a–f, 5a–g, 6a–e, 7c–h and Supplementary Figs. 1a–e, 2c–e, 3a–d, 4a–c) were performed on a setup built around a Nikon Ti2 inverted microscope equipped with a Yokogawa CSU-W1 confocal spinning head, a Plan-Apo objective (100×1.45-NA), a back-illuminated sCMOS camera (Prime 95B; Photometrics) and a super-resolution module (Live-SR; Gataca Systems) that is based on structured illumination with optical reassignment and image processing. Excitation light was provided by 488-nm/150 mW (Coherent) (for GFP/Alexa 488), 561-nm/100 mW (Coherent) (for mCherry/LipidTOX/Alexa 594/ddFP-R) and 642-nm/110 mW (Vortran) (for iRFP/miRFP/Alexa 647) (power measured at optical fiber end) DPSS laser combiner (iLAS system; Gataca systems), and all image acquisition and processing was controlled by MetaMorph (Molecular Device) software. Images were acquired with exposure times in the 500 msec range. For time-lapse imaging, images were sampled at 1/60 Hz (Fig. 7c–h) or 1/900 Hz (Fig. 2b–d) with exposure times in the 500 msec range.

Total internal reflection fluorescence (TIRF) microscopy (Figs. 1i, j, 3b–e and Supplementary Fig. 5a, b) was performed on a setup built around a Nikon Ti2 inverted microscope equipped with a nHP Apo-TIRF objective (×60 1.49-NA), and a back-illuminated sCMOS camera (Prime 95B; Photometrics). Excitation light was provided by 488-nm/70 mW (for GFP), 561-nm/70 mW (for mCherry), and 647-nm/125 mW (for miRFP) (power measured at optical fiber end) DPSS laser combiner (Nikon LU-NV laser unit), coupled to the motorized TIRF illuminator through an optical fiber cable. Critical angle was maintained at different wavelengths throughout the experiment from the motorized TIRF illuminator. Acquisition was controlled by Nikon NIS-Element software. Images were sampled at 1/60 Hz with exposure times in the 500 msec range.

For live cell imaging of HeLa, U2OS, COS-7, and HEK293T cells, cells were washed twice and incubated with Ca²⁺ containing buffer (140 mM NaCl, 5 mM KCl, 1 mM MgCl₂, 10 mM HEPES, 10 mM glucose, and 2 mM CaCl₂, pH 7.4) before imaging with SDC or TIRF microscope.

For live cell imaging of N/TERT cells, THP-1/macrophages, and iPSC-derived neurons, imaging was carried out in their respective culture medium. All types of microscopy were carried out at 37 °C except for experiments detecting accessible PM cholesterol with purified mCherry-D4H proteins, which were conducted at room temperature.

For immunolabeling of iPSC-derived neurons expressing EGFP-GRAM-W (DIV 33), neurons were fixed with 4% paraformaldehyde (PFA) for 15 min at room temperature, gently washed in phosphate-buffered saline (PBS), and permeabilized with PBS containing 0.1% Triton-X and 3% donkey serum for 1 hr before immunostaining with designated antibodies in the PBS containing 0.1% Triton-X and 1% donkey serum. Primary antibodies were incubated at 4 °C overnight followed by incubation with Alexa Fluor-conjugated secondary antibodies at room temperature for 2 hrs. Fixed cells were imaged by SDC microscopy. Images from maximum projected focal planes are shown.

For detection of accessible PM cholesterol with purified mCherry-D4H proteins in HeLa and N/TERT cells, purified mCherry-D4H proteins (15 μg/ml) were added into culture medium in the case of N/TERT cells and into $Ca^{2+}$ containing buffer (140 mM NaCl, 5 mM KCl, 1 mM $MgCl_2$, 10 mM HEPES, 10 mM glucose, and 2 mM $CaCl_2$, pH 7.4) in the case of HeLa cells for 10 min at room temperature unless otherwise stated. Cells were then washed twice with either the culture medium or the $Ca^{2+}$-containing buffer without mCherry-D4H proteins and immediately imaged by SDC microscopy at room temperature.

For LipidTOX staining, HeLa cells were fixed with 4% PFA for 20 min at room temperature, gently washed in PBS, and incubated in PBS containing 50 mM Glycine for 10 min. Cells were then washed thrice with PBS before staining with LipidTOX (1:200 dilution in PBS) for 1 hr at room temperature. Cells were then imaged by SDC microscopy at room temperature.

### Drug stimulation for imaging
For SDC imaging experiments, drugs were used with the following concentration: 50 μM mevastatin (Santa Cruz); 5 mM methyl-β-cyclodextrin (MCD) (Sigma-Aldrich/Merck); 10% LPDS (Sigma-Aldrich/Merck) added to culture medium without FBS (or BPE in the case of N/TERT cells); 200 μM MCD/Cholesterol complex generated as described previously[124]; 50 μg/ml LDL (STEMCELL Technologies); 20 nM OSW-1 (Cayman Chemical); 10 μM Sandoz 58-035 (SZ58-035) (Sigma-Aldrich/Merck).

For SDC imaging experiments with cells cultured in medium containing mevastatin only, LPDS only, or mevastatin and LPDS (Fig. 4b–f), cells were cultured in their respective culture medium (DMEM and 20% FBS for HeLa, U2OS, HEK293T, and COS-7 cells; RPMI and 10% FBS for THP-1 cells; KSFM supplemented with rEGF and BPE for N/TERT cells) for 24 hrs before being incubated in either control culture medium (as described above), culture medium supplemented with 50 μM mevastatin, culture medium supplemented with 10% LPDS (DMEM and 10% LPDS for HeLa, U2OS, HEK293T, and COS-7 cells; RPMI and 10% LPDS for THP-1 cells; KSFM with rEGF and 10% LPDS for N/TERT cells), or culture medium supplemented with 10% LPDS and 50 μM mevastatin for 16 hrs prior to imaging.

For all time-lapse TIRF imaging experiments with drug stimulation, drugs were added to the cells 10 min after the initiation of the imaging [5 mM MCD was added to the cells in Fig. 1i, j, and 200 nM rapamycin (Sigma-Aldrich/Merck) was added to cells in Fig. 3b–e and Supplementary Fig. 5a, b].

### Image analysis
All images were analyzed off-line using ImageJ (http://fiji.sc/wiki/index.php/Fiji). Quantification of fluorescence signals was performed using Excel (Microsoft) and Prism 8 (GraphPad Software). All data are presented as mean ± SEM. In dot plots, each dot represents value from a single cell with the bar as the mean.

For analysis of the recruitment of EGFP-GRAM$_{1b}$ constructs, mCherry-Lact-C2, iRFP-P4M, and iRFP-PH-PLCδ1, to the PM via SDC microscopy (Figs. 1g, h, 2b–d, 4b–f, 5a, b, e–g, 6b–e and Supplementary Figs. 1b–e, 3b, c, 4a, b), line scan analysis was performed. A line of 5 μm in length was manually drawn across the PM, and fluorescence intensity along the manually drawn line was measured. The peak intensity around the PM region was normalized with the intensity of cytoplasmic region and then plotted for quantification.

For analysis of the binding of the purified recombinant mCherry-D4H proteins to the PM via SDC microscopy (Fig. 5c, d and Supplementary Figs. 3b, d, 4a, c), line scan analysis was performed. A line of 5 μm in length was manually drawn across the PM, and the peak fluorescence intensity value of mCherry-D4H at the PM was measured after background subtraction.

For analysis of the number and size of lipid droplets via SDC microscopy (Supplementary Fig. 2c–e), an arbitrary threshold was applied to each image to segment lipid droplets (pixel size cut-off: 5-infinity). Region of interest (ROI) was manually drawn to cover the entirety of individual cells. The number of lipid droplet and the average size of lipid droplets within each cell were then obtained and plotted.

For time-lapse imaging via TIRF microscopy (Fig. 3b–e), changes in PM fluorescence over time were analyzed by manually selecting ROI covering the largest possible area of the cell foot-print. Mean fluorescence intensity values of the selected regions were obtained and normalized to the average fluorescence intensity before stimulation after background subtraction.

For analysis of the ddFP-R-GRAM-H or ddFP-R-GRAM-W fluorescence at PM via SDC microscopy (Fig. 7c–h), line scan analysis was performed. A line of 5 μm in length was manually drawn across the PM, and ddFP fluorescence intensity along manually drawn line was measured. The peak intensity around the PM region was obtained after subtraction of background cytoplasmic signals. It was then normalized by the value at 5 min time point and plotted.

### Flow cytometry
HeLa cells stably expressing PM-RA and B-GRAM-W (ddFP-R-GRAM-W system) or PM-RA and B-GRAM-H (ddFP-R-GRAM-H system) were incubated in control medium or in medium supplemented with either 5 mM MCD for 15 min, 50 μM mevastatin and 10% LPDS for 16 hrs, or 200 μM MCD/Cholesterol for 15 min at 37 °C. Cells were then detached from dishes in PBS containing 5 mM EDTA and harvested by centrifugation. The cells were washed and suspended in PBS, and 10,000 cells were analyzed for each experiment with BD LSRFortessa X-20 (BD Biosciences) to measure fluorescence of ddFP-R-GRAM-W or ddFP-R-GRAM-H, using BD FACSDiva software (BD Biosciences). Data were then analyzed by FlowJo™ software (BD Biosciences) and flow cytometry profile as well as the mean fluorescence intensity for each condition was plotted (Fig. 7i–n).

### Biochemical analyses
**Protein purification.** Protein purification was carried out as previously described[57,68]. All proteins were overexpressed in *E. coli* BL21-DE3 Rosetta cells. A 750 mL culture was grown until $OD_{600}$ ~ 0.5-0.7 with appropriate antibiotics. 0.1 mM IPTG (Thermo Fisher Scientific) was then added, and the culture was further grown at 18 °C for 18 hrs to allow protein expression. Cells were harvested by centrifugation at 4,700 g, at 4 °C for 15 min, and re-suspended in 30 ml of lysis buffer (100 mM HEPES, 500 mM NaCl, 10 mM imidazole, 10% glycerol, 0.5 mM TCEP, pH 7.5), supplemented with protease inhibitors (Complete, EDTA-free; Roche) together with the cocktail of 100 μg/ml lysozyme (Sigma-Aldrich/Merck) and 50 μg/ml DNAse I (Sigma-Aldrich/Merck). Bacteria were lysed with sonication on ice in a Vibra Cell (Sonics and Materials, Inc) (70% power, 3 s pulse on, 3 s pulse off for 3 min for three to five rounds). The lysate was clarified by

centrifugation at 47,000 g, at 4 °C for 20 min. The supernatants were incubated at 4 °C for 30 min with Ni-NTA resin (Thermo Fisher Scientific), which had been equilibrated with 2.5 ml of wash buffer 1 (20 mM HEPES, 500 mM NaCl, 10 mM Imidazole, 10% glycerol, 0.5 mM TCEP, pH 7.5). The protein-resin mixtures were then loaded onto a column to be allowed to drain by gravity. The column was washed with 10 ml of wash buffer 1 twice and 10 ml of wash buffer 2 (20 mM HEPES, 500 mM NaCl, 25 mM imidazole, 10% glycerol, 0.5 mM TCEP, pH 7.5) twice, and then eluted with 1.25 ml of elution buffer 1 (20 mM HEPES, 500 mM NaCl, 500 mM imidazole, 10% glycerol, 0.5 mM TCEP, pH 7.5). The proteins were then concentrated using Vivaspin 20 MWCO 10 kDa or MWCO 30 kDa (GE Healthcare) and further purified by gel filtration (Superdex 200 increase 10/300 GL, GE Healthcare) with elution buffer 2 (20 mM HEPES, 300 mM NaCl, 10% glycerol, 0.5 mM TCEP, pH 7.5), using the AKTA Pure system (GE Healthcare). Relevant peaks were pooled, and the protein sample was concentrated.

### Liposome-based experiments

**Liposome preparation.** Liposomes were prepared as previously described[57,68]. Lipids in chloroform were dried under a stream of $N_2$ gas, followed by further drying in the vacuum for 2 hrs. The dried lipid films were hydrated with HK buffer containing sucrose to form heavy liposomes (0.75 M sucrose, 50 mM HEPES, 120 mM potassium acetate, pH 7.5). Liposomes were then formed by five freeze-thaw cycles (liquid $N_2$ and 37 °C water bath) followed by extrusion using Nanosizer with a pore size of 100 nm (T&T Scientific Corporation).

**Liposome sedimentation assays.** Liposome sedimentation assays were performed as previously described[57,68]. Heavy liposomes were prepared by hydrating 1.6 mM dried lipid films in HK buffer containing 0.75 M sucrose and subjected to freeze-thaw cycles five times. 160 µl of heavy liposomes were pelleted and washed with HK buffer without sucrose twice to remove unencapsulated sucrose. Pelleted heavy liposomes were resuspended in 160 µl HK buffer and incubated with 6.4 µg of the indicated GRAM$_{1b}$ proteins for 1 hr at room temperature. Unbound proteins (supernatant) were separated from liposome-bound proteins (pellet) by centrifugation at 21,000 g for 1 hr at 25 °C. After centrifugation, the supernatant was removed, and pellets were re-suspended in 160 µl HK buffer. 20 µl samples were taken from both fractions and run on SDS-PAGE followed by colloidal blue staining. The protein bands were quantified using Fiji, and the % bound GRAM$_{1b}$ was plotted for each condition.

**Immunoblotting.** Immunoblotting was performed as previously described[57]. HeLa cells or N/TERT cells were lysed in SDS lysis buffer (2% SDS, 150 mM NaCl, 10 mM Tris, pH 8.0). The lysates were treated with Benzonase Nuclease (SantaCruz) and incubated at 60 °C for 20 min followed by incubation at 70 °C for 10 min. Protein concentration of cell lysates was then measured using the bicinchoninic acid assay (BCA assay) kit (Thermo Fisher Scientific) and processed for SDS-PAGE and immunoblotting with standard procedure. All immunoblots were developed by chemiluminescence using the SuperSignal West Dura reagents (Thermo Fisher Scientific) and imaged using Gel-Doc (Bio-Rad).

### Molecular modeling

The modeled structure of the GRAM domain of GRAMD1b was adapted and modified from ref. 68.

### Statistics and reproducibility

No statistical method was used to predetermine sample size, and the experiments were not randomized for live cell imaging. Sample size and information about replicates are described in the figure legends. All experiments were independently conducted at least 2 times to confirm reproducibility. The number of biological replicates for all cell-based experiments and the number of technical replicates for all other biochemical assays are shown as the number of independent experiments within figure legends for each figure. Comparisons of data were carried out by the two-tailed unpaired Student's *t* test, or the one-way ANOVA followed by corrections for multiple comparisons as appropriate with Prism 8.0.1 (GraphPad software). Unless $P < 0.0001$, exact $P$ values are shown within figure legends for each figure. $P > 0.05$ was considered not significant. All data are presented as mean ± SEM unless otherwise noted.

### Reporting summary

Further information on research design is available in the Nature Portfolio Reporting Summary linked to this article.

## Data availability

The authors declare that the data supporting the findings of this study are available within the paper and its supplementary information file. Source Data for Figs. 1b–f, h–j, 2c, d, 3b–e, 4b–f, 5b, d, g, 6c, e, 7e–n, and Supplementary Figs. 1c, e, 2a, b, d, e, 3b, d, 4b, c, 6a are provided with this paper. Reagents and cell lines generated for this study are available directly from the authors upon request within 1–2 months.

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

## Acknowledgements

We thank Raihanah Harion and Haoning Yang for the discussion. This work was supported by the Singapore Ministry of Education Academic Research Fund Tier 2 (MOE-T2EP30120-0002 & MOE2017-T2-2-001), the Singapore Ministry of Education Academic Research Fund Tier 1 (RG20/21), and Grant-in-Aid for Scientific Research (B) (22H02620) from the Japan Society for Promotion of Science (JSPS), and a Nanyang Assistant Professorship (NAP) to Y.S., LKCMedicine LEARN grant to T.N., National Research Foundation Fellowship, Singapore (NRF-NRFF11-2019-0006) and Nanyang Assistant Professorship (NAP) to F.L.Z., MOH-000207, NMRC-LCG SPARKII to K.L.L. T.N. was supported by a fellowship from the JSPS.

## Author contributions

All authors participated in the design of experiments, data analysis, and interpretation. D.H.Z.K. and Y.S. participated in designing the liposome-based assays that were performed by D.H.Z.K. T.N., D.H.Z.K., and Y.S. participated in designing the imaging and cell-based biochemical assays that were performed by D.H.Z.K. and M.N. D.H.Z.K., T.N., and M.N. performed all the genetic manipulations. D.H.Z.K. designed and performed all the protein purification work. D.H.Z.K. and T.N. designed and performed lentivirus production and viral transduction. P.R. and F.L.Z. helped to generate N/TERT stable cell lines expressing EGFP-GRAM-W and provided inputs for experiments related to keratinocytes. Y.J.Y. and K.L.L. provided the iPSC-derived dopaminergic neurons and provided inputs for experiments related to neurons. N.M. performed FACS analysis with the help of T.N. Y.S. and D.H.Z.K. wrote the manuscript with inputs from all the authors.

## Competing interests

The authors declare no competing interests.
