## [Peer Review File · Nature Communications]

REVIEWER COMMENTS

Reviewer #1 (Remarks to the Author):

NCOMMS-23-27967-T

Koh et al.

This manuscript describes the development and characterization of a novel cholesterol biosensor based on the cholesterol-sensing domain of GRAMD1b. The novel biosensor is used to detect the distribution of accessible cholesterol in different membranes in a variety of cell types. Among the observations made is the requirement of OSBP for maintenance of the accessible cholesterol pool and the differential dependence of cell types on exogenous vs. de novo cholesterol sources. Finally, the authors describe a dimerization-based variation of the biosensor that may enable even higher sensitivity for detection of the PM accessible cholesterol pool.

The novelty for this study is the development of a new class of cell-expressible biosensors that would appear to have advantages over earlier cholesterol monitoring tools (e.g. PFO and its derivatives). The data quality is high and evidence is provided to suggest the sensors may have the potential to monitor intracellular cholesterol dynamics, which would be an advance for the field. Enthusiasm for the study, however, is tempered by two significant limitations. First, the characterization relies almost exclusively on immunofluorescence imaging to validate the sensors and does not effectively utilize biochemical methods to corroborate findings. Second, the authors have not adequately demonstrated that the probes are inert and do not alter cholesterol dynamics. Addressing these points is needed to provide a critical assessment of the validity of the biosensors.

Specific points:

1. The Introduction could be strengthened to provide a general audience with a clearer understanding of the concept of accessible cholesterol. As there is no difference in the molecular identity of accessible vs. non-accessible cholesterol, what does “chemically active” cholesterol actually mean with respect to PM pools? There is a strong body of evidence that demonstrates this concept is best understood in terms of membrane biophysics. Mesmin et al. (BBA 2009) developed an elegant model that proposed that accessible cholesterol could be understood in terms of its interaction with membrane phospholipids and the effect of differential acyl chain saturation. Olsen et al. (Biophys J 2013) further updated the model by showing that accessibility of cholesterol is driven not by saturation of cholesterol-phospholipid interactions but rather by bulk membrane remodeling that increases solvent exposure of membrane cholesterol. In the case of the GRAMD-W sensor, it is very likely that the incremental addition of

membrane cholesterol results in accessibility of the cholesterol steroid A ring, as is the case for PFO-D4. Anchoring the concept of accessible cholesterol within a biophysical context would help users of these tools to more effectively interpret data generated by the biosensors.

2. The sole reliance on SMase exposure, a pharmacological treatment, and SREBP processing to probe PM to ER cholesterol movement is a weakness (Fig 2). A more physiological and direct approach to monitor rapid transport of cholesterol from PM to ER would be the addition of exogenous cholesterol (e.g. MBCD-cholesterol complexes) to expand the PM cholesterol pool and then to follow cholesterol esterification and lipid droplet formation (e.g. use of lipidtox).

3. A key question not adequately addressed is whether expression of the biosensors perturbs cholesterol homeostasis. This is hinted at in Fig 3, when comparing Movie S2 (WT) with Movie S3 (5P). Whereas WT IF is characterized by large patchy staining, the 5P staining is distinctly more granular and evenly distributed over the entire membrane. This raises concern that the binding of the GRAMD-W sensor to PM accessible cholesterol is altering membrane cholesterol content or distribution. Orthogonal biochemical assays should be performed to assess cholesterol content, distribution, and trafficking between organellar membranes.

4. The reliance on PM/cytoplasmic ratio to assess PM staining under differential cholesterol conditions (Fig 4) is misleading. For example, in U2OS cells cultured under LPDS conditions, the PM/cytoplasmic EGFP staining is reported in the plot as only ~50% of control conditions when it the IF clearly shows enhanced PM staining under LPDS conditions. It is likely that a confounder here is that the sensor is detecting the increased de novo cholesterol synthesis in the ER (as a result of cholesterol deprivation), and this increase in staining is adventitiously lowering the ratio. An alternate way to normalize PM cholesterol is needed to support conclusions about the cell's reliance on exogenous vs. endogenous cholesterol.

5. Direct comparison of GRAM-W and GRAM-H under cholesterol loading and cholesterol extraction conditions (Fig 7C and D) to reach conclusions regarding the relative performance of the probes.

Reviewer #2 (Remarks to the Author):

This manuscript described the development of a novel genetically encoded lipid biosensor for cholesterol. The paper starts with a systematic screen of mutants in a key cholesterol sensing residue in the GRAM domain of GRAM1b. It identifies a previously described mutant with slightly increased

cholesterol affinity (GRAM-H) and a high affinity mutant with greatly enhanced binding to both cholesterol and PS (GRAM-W). Using a variety of pharmacologic and chemigenetic approaches, GRAM-W targeting to the PM and lysosomes is shown to depend on the presence of cholesterol in a variety of cultured cell types, including iPSC-derived dopaminergic neurons. The requirements for cholesterol synthesis vs import from the serum is also evident using GRAM-W. The authors also use the probe to present evidence for a requirement for OSBP-mediated cholesterol transport at the TGN as being required for PM cholesterol enrichment, which resolves a contentious issue in the field. Lastly, an intensitometric assay is developed at the PM for sensitive detection of cholesterol in cell populations.

Overall, the manuscript is clearly and concisely written, and the data presented have exemplary clarity and quality. This probe is clearly a large improvement in terms of stability, clarity and sensitivity over the state-of-the-art, and we predict it will rapidly take-over as the biosensor of choice for the majority of experiments, despite the author's humble claims that it is additive to these. Because cholesterol biology is so fundamental to the fields of membrane biophysics, cell biology, metabolism and neurodegeneration, this paper will have significant cross-disciplinary impact and is appropriate for publication in a high-profile, cross-disciplinary journal like Nature Communications. Overall, we have only comparatively minor clarifications:

#1 selectivity of the sensor: GRAM-W still shows comparable binding to PS; it is therefore a concern that although GRAM-W absolutely requires cholesterol for membrane binding, this may not be sufficient in cells. Given the probe sees cholesterol in the PM and lysosomes, which have significant cytosolic leaflet PS accumulation, this raises the concern that perhaps cholesterol is not sufficient for binding, and hence pools in e.g. the ER may be invisible to the probe. The reviewers cannot conceive of a feasible experiment to test this, and it is certainly an observation far above the level of scrutiny that previous sensors had (e.g. D4H). However, perhaps this potential caveat can be discussed - i.e., GRAM-W's localization is clear evidence for accessible cholesterol in a membrane, but in some contexts, GRAM-W's absence may not indicate the absence of cholesterol.

#2 We do not see a clear reason to split figures 4 and S4; these would be more informative if presented as a single figure.

#3 in figure 7, the distinction between sensitivity of GRAM-W for cholesterol depletion vs GRAM-H for elevations is well taken. However, this distinction would be much more obvious if the reciprocal experiments were included - i.e. how does cholesterol enhancement change ddFP-GRAM-W signal, and how does cholesterol depletion change GRAM-H. We think that including this comparison will take home the need to select between GRAM-H and -W for specific experiments. Otherwise, its possible most groups will select GRAM-W as it looks like a cleaner localization, despite its inferior sensitivity compared to GRAM-H for some assays.

Reviewer #3 (Remarks to the Author):

Overall, this is an interesting paper that describes the development of a novel sensor for “accessible cholesterol” in the cytoplasmic leaflet of the plasma membrane. This is a very valuable contribution that complements the development of sensors such as ALO-D4 for sensing accessible cholesterol in the exofacial leaflet of the plasma membrane.

While the paper is in general well written and clear, I did have some concerns about the current manuscript that could be dealt with by the authors.

1. The descriptions of accessible cholesterol are in some cases misleading. In the graphical abstract and in Figure 7 accessible cholesterol is illustrated projecting beyond the headgroups of the sphingolipids and phospholipids. There is no evidence for this. More likely based on model membrane studies, there is a concentration-dependent pool of cholesterol that is not well protected by the lipid headgroups.

Related to this, the first sentence of the Discussion states that “Cells transport a biochemically distinct fraction of cholesterol, called accessible cholesterol...” This is incorrect and will be misleading for many readers. Cholesterol in membranes is in a dynamic equilibrium of interactions with its neighboring lipids and proteins. As cholesterol levels rise, the fraction of cholesterol that is unshielded at any instant rises, and this can be bound by proteins such as ALO-D4 or GRAM-W.

2. The response to added LDL seems slow. Internalized LDL is significantly degraded within 1-2 hours. In the study by Das et al cited in the references, ALO-D4 labeling of exofacial increases more rapidly. Given the fast rate of flipping of cholesterol in membranes, it seems likely that the accessible cholesterol will increase simultaneously in both leaflets. It would be a nice validation of the new probe to compare the timing of increases of GRAM-W with increases in the well-characterized ALO-D4 binding in the same cells at least in a few experiments.

3. I think it is important for the authors to emphasize that GRAM-W depends on the presence of anionic lipids as well as accessible cholesterol. Lack of binding in some organelles may not be reflective of levels of accessible cholesterol.

REBUTTAL TO THE COMMENTS RAISED BY THE REVIEWERS

We thank the editor and the reviewers for their constructive comments and suggestions. We have carried out extensive new experimentation to address all the concerns and made necessary changes to the texts.

We have now

- 1) provided additional data to compare ddFP-R-GRAM-H and ddFP-R-GRAM-W systems (for Reviewers #1 and #2).
- 2) investigated the impacts of the expression of GRAM-W on cholesterol homeostasis and shown that the formation of lipid droplets upon cholesterol loading to the plasma membrane (PM) (i.e., a process which requires the transport of cholesterol from the PM to the ER) is not affected by expression of EGFP-GRAM-W in HeLa cells (for Reviewer #1).
- 3) examined different pools of PM cholesterol via well-established cholesterol biosensors (i.e., purified mCherry-tagged D4H proteins for accessible cholesterol and purified mCherry-tagged OlyA proteins for inaccessible cholesterol) and shown that there are no changes in the binding of these biosensors to the PM between HeLa cells expressing EGFP-GRAM-W and control HeLa cells (for Reviewer #1).
- 4) discussed more clearly the requirement for anionic lipids in the discussion (for Reviewers #2 and #3).
- 5) answered to the questions regarding the accuracy of the approach we used to normalize the PM signals for the quantification of the levels of accessible PM cholesterol (for Reviewer #1).
- 6) compared the timing of the increase of the binding of EGFP-GRAM-W to the cytosolic leaflet of the PM with the increase the binding of purified mCherry-tagged D4H proteins to the extracellular leaflet of the PM in pre-starved cells during LDL supplementation (for Reviewer #3)

A specific list of changes is indicated below, followed by a point-by-point rebuttal from **page 2**.

Supplementary Figure 4 is now combined with **Figure 4**.

Supplementary Figure 3 is new, and **old Supplementary Figure 3** is now in **Supplementary Figure 4**

We now have seven main figures (**Fig. 1-7**) and four supplementary figures (**Supplementary Fig. 1-5**).

The following data are new:

Comparison of ddFP-R-GRAM-H and ddFP-R-GRAM-W systems

- **Fig. 7K** (showing the fluorescence signals of ddFP-R-GRAM-W system before and after cholesterol loading)
- **Fig. 7L, M** (showing the fluorescence signals of ddFP-R-GRAM-H system before and after cholesterol depletion)

Impacts of GRAM-W expression on cholesterol homeostasis

- **Supplementary Fig. 2C-E** (showing that the formation of lipid droplets upon cholesterol loading to the PM is not affected by the expression of EGFP-GRAM-W in HeLa cells)
- **Appendix Fig. 1** for Reviewer #1 (showing that the binding of purified mCherry-D4H proteins and purified mCherry-OlyA proteins to the PM is not affected by the expression of EGFP-GRAM-W in HeLa cells)
- **Supplementary Fig. 3** (showing that trapping accessible cholesterol via purified mCherry-D4H proteins results in dissociation of EGFP-GRAM-W from the PM)

Additional quantification of the PM binding of EGFP-GRAM-W in U2OS cells using total internal reflection fluorescence (TIRF) microscopy

- **Appendix Fig. 2** for Reviewer #1 (showing that the TIRF-based approach to quantify the PM binding of EGFP-GRAM-W shows similar results to the confocal microscopy-based approach used in our original manuscript)

Comparison of the timing of the increase of GRAM-W binding to the PM with the increase of D4H binding to the PM

- **Supplementary Fig. 2G-I** (showing that the timing of the increase in the binding of EGFP-GRAM-W to the cytosolic leaflet of the PM is similar to that of purified mCherry-D4H proteins to the extracellular leaflet of the PM in pre-starved cells during LDL supplementation)

-Sentences from the reviewers' comments are *in italics*
-Our responses are in blue

Reviewer #1 (Remarks to the Author):

This manuscript describes the development and characterization of a novel cholesterol biosensor based on the cholesterol-sensing domain of GRAMD1b. The novel biosensor is used to detect the distribution of accessible cholesterol in different membranes in a variety of cell types. Among the observations made is the requirement of OSBP for maintenance of the accessible cholesterol pool and the differential dependence of cell types on exogenous vs. de novo cholesterol sources. Finally, the authors describe a dimerization-based variation of the biosensor that may enable even higher sensitivity for detection of the PM accessible cholesterol pool.

The novelty for this study is the development of a new class of cell-expressible biosensors that would appear to have advantages over earlier cholesterol monitoring tools (e.g. PFO and its derivatives). The data quality is high and evidence is provided to suggest the sensors may have the potential to monitor intracellular cholesterol dynamics, which would be an advance for the field.

We would like to express our gratitude to the reviewer for the positive comment.

Enthusiasm for the study, however, is tempered by two significant limitations. First, the characterization relies almost exclusively on immunofluorescence imaging to validate the sensors and does not effectively utilize biochemical methods to corroborate findings. Second, the authors have not adequately demonstrated that the probes are inert and do not alter cholesterol dynamics. Addressing these points is needed to provide a critical assessment of the validity of the biosensors.

Reply: We thank the reviewer for all the constructive comments and suggestions to further improve our manuscript. To clarify, we expressed and observed our biosensors in live cells. Images were then taken via fluorescence microscopy from non-fixed cells, unlike immunofluorescence, where we usually stain fixed cells with appropriate antibodies. Hence, one of the advantages of our GRAM domain-based biosensors is the detection of accessible cholesterol in live cells in real-time.

Specific points:

1. The Introduction could be strengthened to provide a general audience with a clearer understanding of the concept of accessible cholesterol. As there is no difference in the molecular identity of accessible vs. non-accessible cholesterol, what does “chemically active” cholesterol actually mean with respect to PM pools? There is a strong body of evidence that demonstrates this concept is best understood in terms of membrane biophysics. Mesmin et al. (BBA 2009) developed an elegant model that proposed that accessible cholesterol could be understood in terms of its interaction with membrane phospholipids and the effect of differential acyl chain saturation. Olsen et al. (Biophys J 2013) further updated the model by showing that accessibility of cholesterol is driven not by saturation of cholesterol-phospholipid interactions but rather by bulk membrane remodeling that increases solvent exposure of membrane cholesterol. In the case of the GRAMD-W sensor, it is very likely that the incremental addition of membrane cholesterol results in accessibility of the cholesterol steroid A ring, as is the case for PFO-D4. Anchoring the concept of accessible cholesterol within a biophysical context would help users of these tools to more effectively interpret data generated by the biosensors.

Reply: We thank the reviewer for the detailed information on various models of cholesterol accessibility. We realize that our original introduction did not convey the necessary information of accessible cholesterol to a general audience. We now cited the two papers suggested by the reviewer and revised the relevant section of the introduction as shown below (**Page 3**).

- **Original sentences:** “At steady state, only a small fraction of membrane cholesterol is chemically active, or “accessible” via LTP-mediated non-vesicular transport (Chakrabarti et al., 2017; Das et al., 2014; Gay et al., 2015; Lange et al., 2013; Lange et al., 2004; McConnell & Radhakrishnan, 2003; Ohvo-Rekila et al., 2002; Radhakrishnan & McConnell, 2000; Sokolov & Radhakrishnan, 2010).”
- **New sentences:** “Availability of membrane cholesterol for LTP-mediated non-vesicular transport is tightly regulated by the dynamic interactions between cholesterol and the membrane environment (Huang & Feigenson, 1999; Ikonen, 2008; Mesmin & Maxfield, 2009; Olsen et al., 2013; Radhakrishnan &

McConnell, 2000). At steady state, the majority of membrane cholesterol is inaccessible to LTPs due to sequestration via its complex formation with neighboring membrane lipids, such as phospholipids and sphingomyelin. Thus, only a small fraction of membrane cholesterol is accessible (also known as chemically active) for LTP-mediated non-vesicular transport at steady state (Chakrabarti et al., 2017; Das et al., 2014; Gay et al., 2015; Lange et al., 2013; Lange et al., 2004; McConnell & Radhakrishnan, 2003; Ohvo-Rekila et al., 2002; Radhakrishnan & McConnell, 2000; Sokolov & Radhakrishnan, 2010).”

2. *The sole reliance on SMase exposure, a pharmacological treatment, and SREBP processing to probe PM to ER cholesterol movement is a weakness (Fig 2). A more physiological and direct approach to monitor rapid transport of cholesterol from PM to ER would be the addition of exogenous cholesterol (e.g. MBCD-cholesterol complexes) to expand the PM cholesterol pool and then to follow cholesterol esterification and lipid droplet formation (e.g. use of lipidtox).*

Reply: We thank the reviewer for this constructive suggestion. Being inspired by the reviewer’s suggestion, we performed additional experiments to monitor the formation of lipid droplet (via staining with LipidTOX) upon cholesterol loading to the PM [via incubation of cells with the complexes of cholesterol and methyl- β -cyclodextrin (MCD)] in control HeLa cells and HeLa cells stably expressing EGFP-GRAM-W biosensor. In the revised manuscript, we show that cholesterol loading to the PM induces the formation of lipid droplets in HeLa cells and that such lipid droplet formation is dependent on cholesterol esterification in the ER [as the inhibition of Acyl-CoA:cholesterol acyltransferase (ACAT) by ACAT inhibitor, SZ58-035, resulted in the suppression of the formation of lipid droplets]. Importantly, there were no differences in the number or the size of lipid droplets between control wild-type HeLa cells and HeLa cells stably expressing EGFP-GRAM-W biosensor. These new results are now included in the revised manuscript (**new Supplementary Figure 2C-E**). Our new results further support that the expression of GRAM-W biosensor in cells does not disrupt the movement of accessible cholesterol from the PM to the ER.

3. *A key question not adequately addressed is whether expression of the biosensors perturbs cholesterol homeostasis. This is hinted at in Fig 3, when comparing Movie S2 (WT) with Movie S3 (5P). Whereas WT IF is characterized by large patchy staining, the 5P staining is distinctly more granular and evenly distributed over the entire membrane. This raises concern that the binding of the GRAMD-W sensor to PM accessible cholesterol is altering membrane cholesterol content or distribution. Orthogonal biochemical assays should be performed to assess cholesterol content, distribution, and trafficking between organellar membranes.*

Reply: We would like to provide the reviewer with clarification on the relevant data. We think the reviewer is referring to PM signals of miRFP-FKBP-GRAMD1b WT (**Movie S2**) and miRFP-FKBP-GRAMD1b 5P (**Movie S3**) that were monitored by total internal reflection fluorescence (TIRF) microscopy. In these experiments, PM recruitment of FKBP-tagged GRAMD1b lacking the GRAM domain [either with wild-type (WT) StART-like domain or with the version of the StART-like domain carrying 5P mutations that block its ability to transport cholesterol] was artificially induced via rapamycin-dependent heterodimerization of FKBP and PM-anchored FRB (as illustrated in **Figure 3A**). In this method, the efficiency of the rapamycin-dependent recruitment of FKBP-tagged proteins to the PM vary between individual cells. We acknowledge that the representative image and movie shown in the Figures might have led to confusion. We now replaced the image/movie in **Supplementary Figure 4A** (used to be **Supplementary Figure 3A** in our original manuscript) and **Movie S2** to show a representative condition, where similar levels of miRFP-FKBP-GRAMD1b WT and miRFP-FKBP-GRAMD1b 5P are recruited to the PM.

In addition, we performed orthogonal assays to assess the impacts of the expression of EGFP-GRAM-W on the accessibility of cholesterol in the PM. In the first set of experiments, we incubated cells with purified recombinant mCherry-tagged D4H (mCherry-D4H) protein, a well-established accessible cholesterol biosensor, for the detection of accessible cholesterol in the PM. We found no detectable differences in the binding of mCherry-D4H proteins to the PM between control HeLa cells and HeLa cells expressing EGFP-GRAM-W, as assessed via immunoblotting of total cell lysates and fluorescence microscopy of live cells. In the second set of experiments, we incubated cells with purified recombinant mCherry-tagged OlyA (mCherry-OlyA) protein, an inaccessible cholesterol biosensor that binds to sphingomyelin forming a complex with cholesterol (PMID: 30712872; PMID: 33712199), for the detection of inaccessible cholesterol in the PM. We found no detectable differences in the binding of mCherry-OlyA proteins to the PM between control HeLa cells and HeLa cells expressing EGFP-GRAM-W, as assessed via immunoblotting of total cell lysates and fluorescence microscopy of live cells. These results (shown below in **Appendix Fig. 1**), together with our new results confirming that there is no significant perturbation in the movement of accessible cholesterol from the PM to the ER (**new Supplementary Figure 2C-E**), show that the expression of GRAM-W biosensor in

cells does not cause major disruptions in cholesterol distribution and homeostasis. Furthermore, as shown in **Appendix Fig. 1**, incubation of cells with purified mCherry-D4H proteins for 30 min resulted in increased binding of mCherry-D4H proteins to the PM and the dissociation of EGFP-GRAM-W from the PM (also in **new Supplementary Figure S3A-C**). This is consistent with the property of D4H to trap accessible cholesterol in the extracellular leaflet of the PM (e.g., PMID: 28414269; PMID: 33604931) and reduce accessible cholesterol in the cytosolic leaflet of the PM.

Appendix Figure 1. Expression of GRAM-W does not disrupt the accessibility of PM cholesterol. (A-B) Lysates of wild-type HeLa cells (Control) and HeLa cells stably expressing EGFP-GRAM-W (GRAM-W) that were treated either with purified mCherry-D4H protein (15 μ g/ml) for 30 min at room temperature (A) or with purified OlyA-mCherry protein (1 μ M) for 5 min at room temperature (B) were processed by SDS-PAGE and immunoblotted (IB) with anti-GFP, anti-mCherry, and anti-actin antibodies. (C-D) Quantification of the mCherry band signal intensity of either wild-type HeLa cells (Control) or HeLa cells stably expressing EGFP-GRAM-W (GRAM-W) as analyzed in (A) and (B) (n = 3 for each condition; two-tailed unpaired Student's t-test, n.s. denotes not significant). (E) Confocal images of live Control HeLa cells or GRAM-W HeLa cells stained with or without purified mCherry-D4H protein (15 μ g/ml) for 30 min at room temperature or purified OlyA-mCherry protein (1 μ M) for 5 min at room temperature. Insets show at higher magnification the regions indicated by white dashed boxes. White dotted lines are drawn to depict the PM of cells. Scale bars, 10 μ m. (F-G) Quantification of the mCherry-D4H signals (F) or OlyA-mCherry signals (G), as assessed by confocal microscopy and line scan analysis as shown in (E) (mean \pm SEM, n = 20 cells for each condition; data are pooled from two experiments; two-tailed unpaired Student's t-test, n.s. denotes not significant). (H) Quantification of the ratio of PM signals to cytoplasmic signals of EGFP-GRAM-W, as assessed by confocal microscopy and line scan analysis as shown in (E) (mean \pm SEM, n = 20 cells for each condition; data are pooled from two independent experiments; Dunnett's multiple comparisons test, **P < 0.0001. n.s. denotes not significant). Note the dissociation of EGFP-GRAM-W from the PM induced by accessible cholesterol trapping via 30 min incubation with purified mCherry-D4H proteins.

4. The reliance on PM/cytoplasmic ratio to assess PM staining under differential cholesterol conditions (Fig 4) is misleading. For example, in U2OS cells cultured under LPDS conditions, the PM/cytoplasmic EGFP staining is reported in the plot as only ~50% of control conditions when it the IF clearly shows enhanced PM staining under LPDS conditions. It is likely that a confounder here is that the sensor is detecting the increased *de novo* cholesterol synthesis in the ER (as a result of cholesterol deprivation), and this increase in staining is adventitiously lowering the ratio. An alternate way to normalize PM cholesterol is needed to support conclusions about the cell's reliance on exogenous vs. endogenous cholesterol.

Reply: We thank the reviewer for raising this issue. We would like to clarify the experimental and quantification approach used for the data shown in **Figure 4**. We expressed EGFP-GRAM-W in U2OS cells (and other selected cell types, including HEK293T and COS-7) via “transient transfection” of a plasmid encoding EGFP-GRAM-W, using Lipofectamine 2000. With the transient transfection approach, the amount of EGFP-GRAM-W available in each cell cannot be precisely controlled, resulting in cells with different expression levels of EGFP-GRAM-W, even within the same dish/condition. Hence, it is important to normalize the differences in the expression of EGFP-GRAM-W across different cells. For a genetically-encoded biosensor that binds to lipids in the PM, such as the PH domain of PLC δ 1 for PI(4,5)P₂ detection, taking the ratio of PM signals over cytosolic signals is a common normalization approach (e.g., PMID: 23229899; PMID: 22722250). In this particular image from LPDS-treated U2OS cells, the increase in PM signals is due to higher expression of EGFP-GRAM-W in these cells, compared to other cells shown for control and mevastatin-treated conditions. However, we acknowledge that the image was misleading to suggest that there was an increase in PM signals upon LPDS treatment. We have now replaced the image with a more representative image that reflects the reduction of PM signals in LPDS-treated U2OS cells as well as some previous images with more representative images in **Figure 4**. Regarding the ER, we do not have an evidence that EGFP-GRAM-W can detect accessible cholesterol in the ER. The low levels of anionic lipids in the cytosolic leaflet of the ER likely contribute to the inefficiency of EGFP-GRAM-W to detect accessible ER cholesterol; this point is now more clearly discussed in the discussion of the revised manuscript as shown below (**page 13**).

- **New sentence:** “Importantly, the binding of GRAM domain-based accessible cholesterol biosensors to some organelles with low levels of anionic lipids in cytosolic leaflets, such as the ER, are likely inefficient even in the presence of accessible cholesterol.”

In addition, we performed additional assays to assess the levels of PM-bound EGFP-GRAM-W, using total internal reflection fluorescence (TIRF) microscopy. In these experiments, we normalized the PM fluorescence signals obtained from the footprint of U2OS cells transiently expressing EGFP-GRAM-W in the evanescent field (which result from the fraction of the biosensors that are bound to the PM) by the “total” fluorescence signals acquired via epifluorescence microscopy. Compared to control un-treated condition, EGFP-GRAM-W showed reduced PM signals under the LPDS-treated condition. These results (shown below in **Appendix Fig. 2**) are similar to those obtained through our confocal microscopy-based quantification approach (**Figure 4**), demonstrating that our method of quantification, which involves taking the ratio of PM signals to cytosolic signals, is suitable for normalizing PM signals.

Appendix Figure 2. Total internal reflection fluorescence (TIRF) microscopy-based quantification of U2OS cells expressing EGFP-GRAM-W. (A) TIRF images of live U2OS cells expressing EGFP-GRAM-W. Cells were incubated in either control medium or medium supplemented with indicated components [mevastatin (50 μ M); 10% lipoprotein-deficient serum (LPDS)] for 16 hrs before imaging. Upper panel shows the evanescence (TIRF) view while the bottom panel shows the epifluorescence view. Scale bars, 10 μ m. (B) Quantification of the PM-bound EGFP-GRAM-W signals (from TIRF view) normalized by the cytosolic signals of EGFP-GRAM-W (from epifluorescence view) (mean \pm SEM, n = 62 cells (Control), n = 55 cells (Mevastatin), n = 56 (LPDS), n = 57 (Mevastatin + LPDS); data are pooled from two independent experiments; Dunnett's multiple comparisons test, **P < 0.0001. n.s. denotes not significant).

5. *Direct comparison of GRAM-W and GRAM-H under cholesterol loading and cholesterol extraction conditions (Fig 7C and D) to reach conclusions regarding the relative performance of the probes.*

Reply: We acknowledge the reviewer's suggestion regarding the importance of comparing GRAM-W and GRAM-H in the experiments presented in **Figure 7**. We have included the additional conditions (**Figure 7K-M**), as per the reviewer's suggestion. Our results show that cholesterol loading slightly increased ddFP-R-GRAM-W fluorescence signals, while it induces prominent increase in ddFP-R-GRAM-H fluorescence signals (~2 fold increase). On the other hand, cholesterol depletion either via MCD treatment or via the combined treatment with mevastatin and LPDS strongly reduced ddFP-R-GRAM-H fluorescence signals, although overall decrease of the fluorescence signals upon the MCD treatment was generally less for the ddFP-R-GRAM-H system compared to the ddFP-R-GRAM-W system (~4 fold decrease in the case for ddFP-R-GRAM-W fluorescence signals vs. ~3 fold decrease in the case for ddFP-R-GRAM-H fluorescence signals). These data support our recommendation that the GRAM-W is most suitable for detecting the decrease in accessible cholesterol in the PM and that the GRAM-H is most suitable for detecting the increase of accessible cholesterol in the PM. We added the following new sentence in **Page 11**.

- New sentence: "They also show that the ddFP-R-GRAM-W system (and more generally GRAM-W) is most suitable for detecting the decrease in accessible cholesterol in the PM and that the ddFP-R-GRAM-H system (and more generally GRAM-H) is most suitable for detecting the increase of accessible cholesterol in the PM."

Reviewer #2 (Remarks to the Author):

This manuscript described the development of a novel genetically encoded lipid biosensor for cholesterol. The paper starts with a systematic screen of mutants in a key cholesterol sensing residue in the GRAM domain of GRAM1b. It identifies a previously described mutant with slightly increased cholesterol affinity (GRAM-H) and a high affinity mutant with greatly enhanced binding to both cholesterol and PS (GRAM-W). Using a variety of pharmacologic and chemigenetic approaches, GRAM-W targeting to the PM and lysosomes is shown to depend on the presence of cholesterol in a variety of cultured cell types, including iPSC-derived dopaminergic neurons. The requirements for cholesterol synthesis vs import from the serum is also evident using GRAM-W. The authors also use the probe to present evidence for a requirement for OSBP-mediated cholesterol transport at the TGN as being required for PM cholesterol enrichment, which resolves a contentious issue in the field. Lastly, an intensitometric assay is developed at the PM for sensitive detection of cholesterol in cell populations.

Overall, the manuscript is clearly and concisely written, and the data presented have exemplary clarity and quality. This probe is clearly a large improvement in terms of stability, clarity and sensitivity over the state-of-the-art, and we predict it will rapidly take-over as the biosensor of choice for the majority of experiments, despite the author's humble claims that it is additive to these. Because cholesterol biology is so fundamental to the fields of membrane biophysics, cell biology, metabolism and neurodegeneration, this paper will have significant cross-disciplinary impact and is appropriate for publication in a high-profile, cross-disciplinary journal like Nature Communications. Overall, we have only comparatively minor clarifications:

We thank the reviewer for all the positive and constructive comments to further improve our manuscript.

#1 selectivity of the sensor: GRAM-W still shows comparable binding to PS; it is therefore a concern that although GRAM-W absolutely requires cholesterol for membrane binding, this may not be sufficient in cells. Given the probe sees cholesterol in the PM and lysosomes, which have significant cytosolic leaflet PS accumulation, this raises the concern that perhaps cholesterol is not sufficient for binding, and hence pools in e.g. the ER may be invisible to the probe. The reviewers cannot conceive of a feasible experiment to test this, and it is certainly an observation far above the level of scrutiny that previous sensors had (e.g. D4H). However, perhaps this potential caveat can be discussed - i.e., GRAM-W's localization is clear evidence for accessible cholesterol in a membrane, but in some contexts, GRAM-W's absence may not indicate the absence of cholesterol.

Reply: We thank the reviewer for this important suggestion. We agree with the reviewer that the absence of the binding of EGFP-GRAM-W to certain organelles with low levels of anionic lipids does not necessarily mean that accessible cholesterol is absent in those organelles. We inserted the following new sentence in the discussion to discuss this point more clearly (**Page 13**).

- **New sentence:** “Importantly, the binding of GRAM domain-based accessible cholesterol biosensors to some organelles with low levels of anionic lipids in cytosolic leaflets, such as the ER, are likely inefficient even in the presence of accessible cholesterol.”

#2 We do not see a clear reason to split figures 4 and S4; these would be more informative if presented as a single figure.

Reply: We thank the reviewer for the suggestion. We have now combined **Figure 4** and **Supplementary Figure 4** into one figure (**new Figure 4**).

#3 in figure 7, the distinction between sensitivity of GRAM-W for cholesterol depletion vs GRAM-H for elevations is well taken. However, this distinction would be much more obvious if the reciprocal experiments were included - i.e. how does cholesterol enhancement change ddFP-GRAM-W signal, and how does cholesterol depletion change GRAM-H. We think that including this comparison will take home the need to select between GRAM-H and -W for specific experiments. Otherwise, its possible most groups will select GRAM-W as it looks like a cleaner localization, despite its inferior sensitivity compared to GRAM-H for some assays.

Reply: We acknowledge the reviewer's suggestion regarding the importance of comparing GRAM-W and GRAM-H in the experiments presented in **Figure 7**. We have included the additional conditions (**Figure 7K-M**), as per the reviewer's suggestion. Our results show that cholesterol loading slightly increased ddFP-R-

GRAM-W fluorescence signals, while it induces prominent increase in ddFP-R-GRAM-H fluorescence signals (~2 fold increase). On the other hand, cholesterol depletion either via MCD treatment or via the combined treatment with mevastatin and LPDS strongly reduced ddFP-R-GRAM-H fluorescence signals, although overall decrease of the fluorescence signals upon the MCD treatment was generally less for the ddFP-R-GRAM-H system compared to the ddFP-R-GRAM-W system (~4 fold decrease in the case for ddFP-R-GRAM-W fluorescence signals vs. ~3 fold decrease in the case for ddFP-R-GRAM-H fluorescence signals). These data support our recommendation that the GRAM-W is most suitable for detecting the decrease in accessible cholesterol in the PM and that the GRAM-H is most suitable for detecting the increase of accessible cholesterol in the PM. We added the following new sentence in **Page 11**.

- New sentence: “They also show that the ddFP-R-GRAM-W system (and more generally GRAM-W) is most suitable for detecting the decrease in accessible cholesterol in the PM and that the ddFP-R-GRAM-H system (and more generally GRAM-H) is most suitable for detecting the increase of accessible cholesterol in the PM.”

Reviewer #3 (Remarks to the Author):

Overall, this is an interesting paper that describes the development of a novel sensor for “accessible cholesterol” in the cytoplasmic leaflet of the plasma membrane. This is a very valuable contribution that complements the development of sensors such as ALO-D4 for sensing accessible cholesterol in the exofacial leaflet of the plasma membrane.

While the paper is in general well written and clear, I did have some concerns about the current manuscript that could be dealt with by the authors.

We thank the reviewer for all the positive and constructive comments to improve our manuscript.

1. The descriptions of accessible cholesterol are in some cases misleading. In the graphical abstract and in Figure 7 accessible cholesterol is illustrated projecting beyond the headgroups of the sphingolipids and phospholipids. **There is no evidence for this.** More likely based on model membrane studies, there is a concentration-dependent pool of cholesterol that is not well protected by the lipid headgroups.

Reply: We thank the reviewer for the comment. We have edited the graphical abstract and illustration in **Figure 7**. We also realize that our original introduction did not convey the necessary information of accessible cholesterol to a general audience. We now revised the relevant section of the introduction as shown below (**Page 3**).

- **Original sentences:** “At steady state, only a small fraction of membrane cholesterol is chemically active, or “accessible” via LTP-mediated non-vesicular transport (Chakrabarti et al., 2017; Das et al., 2014; Gay et al., 2015; Lange et al., 2013; Lange et al., 2004; McConnell & Radhakrishnan, 2003; Ohvo-Rekila et al., 2002; Radhakrishnan & McConnell, 2000; Sokolov & Radhakrishnan, 2010).”
- **New sentences:** “Availability of membrane cholesterol for LTP-mediated non-vesicular transport is tightly regulated by the dynamic interactions between cholesterol and the membrane environment (Huang & Feigenson, 1999; Ikonen, 2008; Mesmin & Maxfield, 2009; Olsen et al., 2013; Radhakrishnan & McConnell, 2000). At steady state, the majority of membrane cholesterol is inaccessible to LTPs due to sequestration via its complex formation with neighboring membrane lipids, such as phospholipids and sphingomyelin. Thus, only a small fraction of membrane cholesterol is accessible (also known as chemically active) for LTP-mediated non-vesicular transport at steady state (Chakrabarti et al., 2017; Das et al., 2014; Gay et al., 2015; Lange et al., 2013; Lange et al., 2004; McConnell & Radhakrishnan, 2003; Ohvo-Rekila et al., 2002; Radhakrishnan & McConnell, 2000; Sokolov & Radhakrishnan, 2010).”

Related to this, the first sentence of the Discussion states that “Cells transport a biochemically distinct fraction of cholesterol, called accessible cholesterol...” This is incorrect and will be misleading for many readers. Cholesterol in membranes is in a dynamic equilibrium of interactions with its neighboring lipids and proteins. As cholesterol levels rise, the fraction of cholesterol that is unshielded at any instant rises, and this can be bound by proteins such as ALO-D4 or GRAM-W.

Reply: We thank the reviewer for the comment. We now removed the relevant sentence from the discussion to avoid confusion.

2. The response to added LDL seems slow. Internalized LDL is significantly degraded within 1-2 hours. In the study by Das et al cited in the references, ALO-D4 labeling of exofacial increases more rapidly. Given the fast rate of flipping of cholesterol in membranes, it seems likely that the accessible cholesterol will increase simultaneously in both leaflets. It would be a nice validation of the new probe to compare the timing of increases of GRAM-W with increases in the well-characterized ALO-D4 binding in the same cells at least in a few experiments.

Reply: We thank the reviewer for the suggestion. We performed new experiments to simultaneously monitor the binding of purified mCherry-D4H proteins to the extracellular leaflet of the PM and the binding of EGFP-GRAM-W to the cytosolic leaflet of the PM in pre-starved HeLa cells at different time points during LDL supplementation. HeLa cells stably expressing EGFP-GRAM-W were starved using mevastatin and LPDS for 16 hours and supplemented with LDL for different periods of time. They were subsequently incubated with purified recombinant mCherry-D4H protein for 10 min. The binding of mCherry-D4H proteins to the extracellular leaflet of the PM and the binding of EGFP-GRAM-W to the cytosolic leaflet of the PM were then

simultaneously analyzed by SDC microscopy. Overall, the timing of the increase in the binding of EGFP-GRAM-W to the cytosolic leaflet of the PM was similar to that of purified mCherry-D4H proteins to the extracellular leaflet of the PM. We detected an increase in the PM binding of EGFP-GRAM-W and purified mCherry-D4H proteins 4 hours after LDL treatment. The binding of both biosensors increased gradually over time during a 12-hour time course, supporting the occurrence of an increase in accessible cholesterol in both leaflets of the PM upon LDL treatment. We now include the new data in the revised manuscript (**new Supplementary Figure 2G-I**). The difference in the time required to detect the binding of D4-based biosensors between our study and the study by Das et al. (PMID: 24920391) could be due to differences in the cell lines used in the assays; while we used HeLa cells, Das et al. used SV-589 cells.

3. I think it is important for the authors to emphasize that GRAM-W depends on the presence of anionic lipids as well as accessible cholesterol. Lack of binding in some organelles may not be reflective of levels of accessible cholesterol.

Reply: We thank the reviewer for this important suggestion. We agree with the reviewer that the absence of the binding of EGFP-GRAM-W to certain organelles with low levels of anionic lipids does not necessarily mean that accessible cholesterol is absent in those organelles. We inserted the following new sentence in the discussion to discuss this point more clearly (**Page 13**).

- **New sentence:** “Importantly, the binding of GRAM domain-based accessible cholesterol biosensors to some organelles with low levels of anionic lipids in cytosolic leaflets, such as the ER, are likely inefficient even in the presence of accessible cholesterol.”

REVIEWERS' COMMENTS

Reviewer #1 (Remarks to the Author):

The authors have suitably revised the manuscript addressing constructively all of my concerns raised in the initial review. The new data provided in Fig 2C suggests that the expression of the probe does not perturb PM to ER cholesterol movement and that the data in Fig 3A suggests the probes do not alter PM cholesterol organization. While this data increases confidence that the probes may be inert, there are many aspects of cholesterol membrane biology not yet examined. The authors might consider addressing this potential limitation in the Discussion. Overall, the study is well-executed and a nice addition to the field.

Reviewer #2 (Remarks to the Author):

The authors have comprehensively addressed mine and the other reviewer's comments. I believe the manuscript is ready for publication.

Reviewer #3 (Remarks to the Author):

The authors have addressed all my concerns in the initial review.

REBUTTAL TO THE COMMENTS RAISED BY THE REVIEWERS

We thank the editor and the reviewers for their constructive comments and suggestions. We have made necessary changes to the texts.

We have now

- 1) discussed potential limitations of the GRAM-W in the discussion

A specific list of changes is indicated below, followed by a point-by-point rebuttal from **page 2**.

-Sentences from the reviewers' comments are *in italics*
-Our responses are in blue

Reviewer #1 (Remarks to the Author):

The authors have suitably revised the manuscript addressing constructively all of my concerns raised in the initial review. The new data provided in Fig 2C suggests that the expression of the probe does not perturb PM to ER cholesterol movement and that the data in Fig 3A suggests the probes do not alter PM cholesterol organization. While this data increases confidence that the probes may be inert, there are many aspects of cholesterol membrane biology not yet examined. The authors might consider addressing this potential limitation in the Discussion. Overall, the study is well-executed and a nice addition to the field.

We thank the reviewer for all the positive and constructive comments to further improve our manuscript.

Reply: We thank the reviewer for this suggestion. We inserted the following new sentence in the discussion to discuss this potential limitation (**Page 13**).

- New sentence: "While the expression of GRAM-W in our assays did not significantly alter the movement of accessible cholesterol in cells and the organization of accessible cholesterol in the PM, it might potentially affect other aspects of cholesterol membrane biology."

Reviewer #2 (Remarks to the Author):

The authors have comprehensively addressed mine and the other reviewer's comments. I believe the manuscript is ready for publication.

We thank the reviewer for all the positive and constructive comments to improve our manuscript.

Reviewer #3 (Remarks to the Author):

The authors have addressed all my concerns in the initial review.

We thank the reviewer for all the positive and constructive comments to improve our manuscript.